# Live attenuated vaccination protects aged chimeric ACE2 mice from severe SARS-CoV-2 pathogenicity in vivo

Alina Russ[1☯], Vera Viherlehto[1☯], Stefanie Brey[2], Sabine Wittmann[1], Pascal Irrgang[1], Natascha Leicht[3], Arne Cordsmeier[1], Armin Ensser[1], Ralf J. Rieker[3,4], Carol Geppert[3,4], Heinrich Sticht[5], Matthias Tenbusch[1,6], Thomas H. Winkler[2], Thomas Gramberg[1,6]*

**1** Harald zur Hausen Institute of Virology, Uniklinikum Erlangen, Friedrich-Alexander-Universität Erlangen-Nürnberg (FAU), Erlangen, Germany, **2** Department of Biology, Nikolaus-Fiebiger-Center for Molecular Medicine, Friedrich-Alexander-Universität Erlangen-Nürnberg (FAU), Erlangen, Germany, **3** Institute of Pathology, Uniklinikum Erlangen, Friedrich-Alexander-Universität Erlangen-Nürnberg (FAU), Erlangen, Germany, **4** Comprehensive Cancer Center Erlangen-EMN (CCC ER-EMN), Bavarian Cancer Research Center (BZKF), Uniklinikum Erlangen, Friedrich-Alexander-Universität Erlangen-Nürnberg (FAU), Erlangen, Germany, **5** Division of Bioinformatics, Institute of Biochemistry, Friedrich-Alexander-Universität Erlangen-Nürnberg (FAU), Erlangen, Germany, **6** FAU Profile Center Immunomedicine (FAU I-MED), Friedrich-Alexander-Universität Erlangen-Nürnberg (FAU), Erlangen, Germany

☯ These authors contributed equally to this work.

* thomas.gramberg@fau.de

## Abstract

Advanced age is one of the greatest risk factors for a severe outcome of COVID-19. Although mRNA vaccines were highly successful in protecting the elderly, the strongest increase in morbidity and mortality upon infection with emerging SARS-CoV-2 variants was among the elderly. To better understand SARS-CoV-2 pathogenicity and to thoroughly evaluate novel vaccination strategies, better models reliably reproducing human SARS-CoV-2 pathogenicity are needed. Here, we generated mice expressing a human-mouse chimera of ACE2 (chACE2) by CRISPR/Cas9-mediated gene editing in C57BL/6 mouse zygotes. ChACE2 mice express the chimeric viral receptor at physiological levels, enabling efficient SARS-CoV-2 infection without the heightened mortality seen in K18-hACE2 mice due to neuroinvasion. We used the chACE2 model to analyze SARS-CoV-2 infection as well as antiviral immune responses in vitro and in vivo. Similar to SARS-CoV-2 in elderly humans, aged chACE2 mice suffered from a highly aggravated disease. In addition, we found that a live attenuated vaccine candidate, LAV[Nsp16], induces robust mucosal and systemic immune responses in these mice despite being highly attenuated. The immunization with LAV[Nsp16] protected aged chACE2 mice from otherwise severe pathogenicity of SARS-CoV-2 by blocking viral replication of homologous and heterologous SARS-CoV-2 variants. The newly developed chACE2 model allows for longer observation periods of SARS-CoV-2 infection in mice, which is essential for assessing the immunogenicity of novel vaccine designs or monitoring viral pathogenicity over time. Immunization with LAV[Nsp16] induced robust and protective immune responses in

**Data availability statement:** All raw data required to replicate the results of the study are submitted as supporting information.

**Funding:** This work was supported by BMBF SenseCoV2 01KI20172A (TG) and IZKF Erlangen project A95 (SW) and A101 (PI). TW was supported by DFG TRR130, Project number 215346292. Further support was received by the DFG through the research training group RTG 2504 (401821119, VV) and by the Bavarian State Ministry for Science and the Arts via the For-COVID project (MT). The funders had no role in study design, data collection and analysis, decision to publish, or preparation of the manuscript.

**Competing interests:** The authors have declared that no competing interests exist.

young and aged mice, making viruses lacking 2'-O-methyltrasferse activity promising candidates for future live attenuated vaccine development.

---

## Author summary

To date, most studies describing SARS-CoV-2 pathology and antiviral immune responses have been conducted in artificial and insufficient murine models. Thus, there is an urgent need for easily expandable mouse models that accurately reflect SARS-CoV-2 pathogenicity in humans. Hence, we generated a novel murine infection model, chACE2, with physiological expression of a humanized form of the receptor ACE2. We found that chACE2 mice support SARS-CoV-2 infection but do not display the enhanced mortality due to artificial and fatal viral neurodissemination seen in other mouse models. Similar to infections in humans, aged chACE2 animals suffered from severe SARS-CoV-2 pathogenicity. Using the improved chACE2 model, we also found that our previously described live attenuated vaccine candidate LAV[Nsp16] is highly attenuated in vivo, induces local and systemic immune responses, and protects aged mice from reinfection with homologous or heterologous SARS-CoV-2 strains. Together, the here described SARS-CoV-2 infection model, chACE2, supports viral replication without fatal viral neurodissemination. It offers a greatly improved way of analyzing COVID-19 pathogenesis in a mouse model and allows for longitudinal analysis of SARS-CoV-2 infection and vaccination.

## Introduction

During the COVID-19 pandemic caused by the severe respiratory syndrome coronavirus 2 (SARS-CoV-2), vaccines based on various strategies were developed in an unprecedented speed. Next to classical approaches, like protein subunit vaccines or inactivated pathogen vaccines, new vaccination strategies were developed such as viral vector-based vaccines or mRNA-based approaches targeting the Spike (S) protein. [1] Specifically, mRNA vaccines such as mRNA-1273 by Moderna or BNT162b2 (Comirnaty) by BioNTech/Pfizer were highly successful and effective in protecting from severe disease and reducing virus transmission in the early phase. [2,3] Although the pandemic subsided with the appearance and successful spread of the Omicron virus variant and its extensive diversification into sublineages in 2021/2022, problems of vaccine strategies targeting solely the S protein became visible during the course of the pandemic. [4] Immune escape and breakthrough infections by newly emerging variants of concern (VOC) occurred frequently in fully vaccinated individuals and made the timely adaptation of mRNA vaccines and annual booster immunization necessary to ensure protection. [5–7] Live attenuated vaccines (LAV) might be a promising vaccine design to overcome those problems since they provide a broader antiviral response, due to the greater variety of viral antigens delivered,

and might induce an efficient mucosal immune response when administered via an intranasal or intratracheal route. [8] However, to this date, no SARS-CoV-2 vaccine candidate using this strategy has been approved for clinical use. Previously, we generated an attenuated SARS-CoV-2 strain, based on Pangolin B.1, in which the viral 2′-O-methyltransferase gene, NSP16, is catalytically inactivated by two point mutations. [9] We found SARS-CoV-2 lacking Nsp16 to be highly immunogenic, resulting in a strongly enhanced release of type I interferon upon infection in vitro, and to be highly interferon-sensitive at the same time. In detail, Nsp16 proved to be important for avoiding efficient recognition by the RNA sensor MDA5 and for shielding viral RNA from the interferon-induced antiviral protein IFIT1. Thus, we identified Nsp16 as a promising target for generating attenuated and highly immunogenic SARS-CoV-2 variants, which might serve as potential LAV candidates. Our findings are in line with work from the Jin group, who also identified virus lacking Nsp16 as highly immunogenic in vitro and found Delta Nsp16 virus replication to be highly attenuated in hamsters and transgenic mice. [10] They showed that a single dose of Delta Nsp16 administered intranasally results in sterilizing immunity in mice and hamsters and prevents viral spread, confirming the idea of a Delta Nsp16 virus as promising SARS-CoV-2 LAV candidate in humans. Subsequently, the group modified their Delta Nsp16-based LAV candidate by additionally inverting the sequence of ORF3a, thereby further reducing pathogenicity and potentially improving safety of the LAV candidate [11].

Advanced age is the most important risk factor associated with severe COVID-19 outcomes. [12–14] Although, SARS-CoV-2 mRNA vaccination led to a strong decline in hospitalization and fatality rate in elderly patients during the early phases of the pandemic, an increase of severe cases in vaccinated elderly compared to younger vaccinated individuals was observed with the emergence of new SARS-CoV-2 variants such as Omicron. [15] How age modulates SARS-CoV-2 reinfection and vaccine breakthrough infections is not completely understood but immunosenescens, including impaired innate responses, interferon signaling, antigen presentation, CD8+ T cell function and many more, has been suggested to play a role. [16] Using mouse-adapted virus variants, several groups started to analyze the age-dependent severity of SARS-CoV-2 in aged mice. [17–19] Analyzing the immune response upon mRNA vaccination and infection, Chen and colleagues found that the interferon and antibody response was impaired in elderly animals and that aged mice showed increased susceptibility to re-infection due to insufficient immunity acquired during primary infection. [18] Together, these findings call for optimized vaccination strategies against severe COVID-19, especially for elderly individuals. Here, we describe the use of a live attenuated vaccine candidate lacking Nsp16 methyltransferase activity, LAV$^{Nsp16}$, in a newly developed SARS-CoV-2 murine infection model, with mice expressing a humanized, chimeric ACE2 receptor (chACE2) at physiological levels. Using chACE2 mice, we find that the promising LAV candidate LAV$^{Nsp16}$ is highly attenuated, induces local and systemic immune responses, and protects aged mice from reinfection with homologous and heterologous SARS-CoV-2 variants.

## Results

### Attenuated replication and pathogenicity of SARS-CoV-2 LAV lacking Nsp16 in vivo

Previously, we described the attenuated replication of SARS-CoV-2 lacking the 2'-O-methyltransferase activity of Nsp16 in human lung epithelial cell cultures. [9] The reduced replication capacity is especially pronounced in the presence of a type I interferon (IFN) response, making the mutant virus a prime candidate for a live attenuate vaccine design (LAV$^{Nsp16}$). To characterize the replication capacity, immunogenicity, and pathogenicity of LAV$^{Nsp16}$ in vivo, mice are the preferred small animal model due to the existence of a high number of available genetically engineered strains. With the exception of certain variants, such as Beta/ B.1.351 or Omicron/ BA.1.1, however, mice are not susceptible to SARS-CoV-2 infection since the murine ACE2 (mACE2) receptor is not recognized with sufficiently high affinity by the receptor binding domain (RBD) of the SARS-CoV-2 Spike (S) protein. [20] Thus, we first used K18-hACE2 transgenic mice, which express multiple copies of the human receptor ACE2 (hACE2), to test LAV$^{Nsp16}$ in vivo. K18-hACE2 mice strongly express hACE2 mainly in epithelial cells under the control of the cytokeratin 18 (K18) promoter and were originally generated as small animal model for SARS-CoV. [21] We infected K18-hACE2 mice intranasally with either SARS-CoV-2 wildtype (wt) or LAV$^{Nsp16}$ and

monitored viral replication as well as pathogenicity of both viruses over the course of seven days (Fig 1). [22] In line with our in vitro studies, we observed an attenuated replication of LAV[Nsp16] compared to wt virus in bronchoalveolar lavage fluid (BAL) and lung tissue of infected mice as determined by RT-qPCR assessing genomic viral RNA level (Fig 1A,1B). [9]. In addition, we detected high SARS-CoV-2 genomic RNA level in the brain of all SARS-CoV-2 wt-infected and in 50% of LAV[Nsp16]-infected mice (Fig 1C). To assess lung tissue damage and epithelial barrier function in the lung of infected mice, we quantified total protein concentration in the BAL as surrogate marker for tissue damage as reported previously. [23, 24] The neurotropism of SARS-CoV-2 in intranasally infected mice has previously been linked to unphysiological expression of the transgene in K18-hACE2 mice and is believed to result in strongly enhanced mortality upon infection. [25–27] In line with these findings, all wt-infected K18-hACE2 mice showed severe virus-induced pathology and reached endpoint criteria at day five to six postinfection (Fig 1E-G). Compared to wt virus-infected animals, LAV[Nsp16]-infected mice showed a strongly attenuated disease progression and mortality with a 75% survival rate at day seven postinfection (Fig 1E). Viral load in the brain of wt- and LAV[Nsp16]-infected K18-hACE2 mice positively correlated with the upregulation of interferon β1 (*Ifnb1*) transcripts in the brain, as a marker of inflammation, and with a strongly enhanced clinical score of infected mice (S1 Fig).

## Generation of mice expressing chimeric human-murine ACE2 receptor

To longitudinally assess immune responses and pathogenesis upon SARS-CoV-2 infection and LAV[Nsp16] vaccination, we needed to overcome the high mortality rate in K18-hACE2 mice upon infection. Thus, we established a murine SARS-CoV-2 infection model expressing physiological levels of a humanized ACE2 receptor, thereby eliminating the need for virus adaptation to the murine receptor. To preserve endogenous functions of the murine receptor, we aimed at introducing only single human residues into the chimeric receptor. Although the murine and the human receptor are highly similar in sequence, they differ substantially in the binding site to the S protein RBD (Fig 2A-C). To understand the differences between human and murine ACE2, we compared the crystal structure of the RBD-hACE2 complex to a model of an RBD-mACE2 complex (Fig 2A, 2B). [28] We particularly focused on the ACE2 sequence stretch spanning residues 24–34, which contains four differences on the amino acid level between hACE2 and mACE2 (Q24N, D30N, K31N, H34Q). In hACE2, all residues at these four sequence positions form favorable polar interactions with the RBD domain (Fig 2B). In contrast, these polar intermolecular interactions are weaker or completely absent in the mACE2-RBD complex and even a steric clash is observed at the interface (Fig 2A). Our modelling suggests that the sequence differences at position 24, 30, 31, and 34 result in a weaker RBD-binding of mACE2 compared to hACE2. To enhance RBD binding, we introduced four mutations (N24Q, N30D, N31K, Q34H) in mACE2 resulting in a chimeric ACE2 (chACE2) receptor (Fig 2C). We used CRISPR-Cas9 technology to replace the endogenous mACE2 with the chACE2 sequence via microinjection into zygotes to generate C57BL/6-chACE2 knock-in mice. When comparing ACE2 expression in chACE2 mice with wt mice and K18-hACE2 mice, we found drastically lower expression of ACE2 in lung and brain tissue compared to K18-hACE2 mice, but similar levels as in wt C57BL/6 mice (Fig 2D). Since a site-directed mutagenesis approach was used on the murine Ace2 gene, the expression pattern of chimeric ACE2 reflects wt mACE2 expression in the different organs of chACE2 mice. Thus, we successfully generated a novel murine SARS-CoV-2 infection model expressing physiological levels of a human-murine chimeric ACE2 receptor.

## ChACE2 mice support SARS-CoV-2 wt and LAV[Nsp16] replication

Next, we asked whether chACE2 mice support replication of wt virus as well as LAV[Nsp16] and monitored viral replication and pathogenicity in young (8–12 weeks) mice over the course of seven days (Fig 3). To compensate for the hACE2 overexpression in K18-hACE2 mice potentially facilitating a more efficient replication, we increased viral inoculum in the chACE2 model (Fig 1A, 3C). Analyzing BAL and lung tissue, we observed that chACE2 mice efficiently support wt virus and LAV[Nsp16] replication and that both viruses show reduced pathogenicity in chACE2 mice compared to K18-hACE2

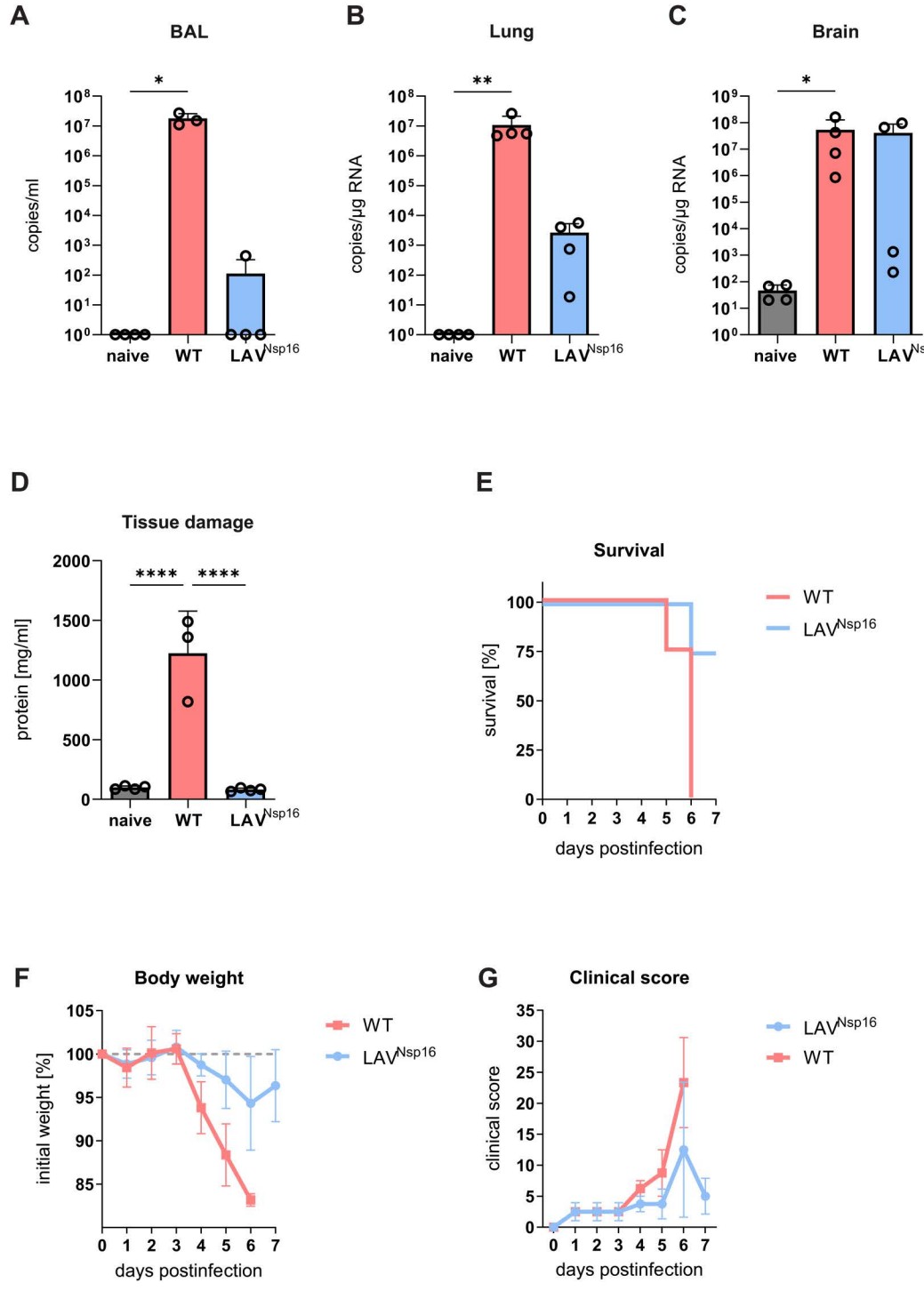

**Fig 1. Attenuated replication of LAV^Nsp16 in K18-hACE2 mice.** B6.Cg-Tg(K18-ACE2)2Prlmn/J mice were infected intranasally with 1,000 PFU of wt or LAV^Nsp16 virus in 30 µl PBS (n = 4). Animals were sacrificed on day seven postinfection, or when predefined welfare endpoints (clinical score ≥ 20) were reached. Viral genome titers in bronchoalveolar lavage (BAL) **(A)**, lung- **(B)** and brain tissue **(C)** were determined by RT-qPCR and are depicted as mean +/- s.d. overlaid with individual data points. Samples with undetectable viral load were set to 1. **(A-C)** Statistical analysis was done using Kruskal-Wallis tests and corrected for multiple comparison using the Dunn's test. **(D)** As a surrogate for tissue damage, protein concentration in the bronchoalveolar lavage fluid was determined by BCA assay. Concentrations are depicted as mean +/- s.d. overlaid with individual data points. Statistical analysis was performed using ordinary one-way ANOVA, followed by Tukey's correction for multiple comparison. **(E)** Survival of wt- or LAV^Nsp16-infected K18-ACE2 mice (n = 4) was monitored over the course of seven days. **(F)** Weight of wt and LAV^Nsp16 infected animals (n = 4) is normalized to the initial bodyweight.

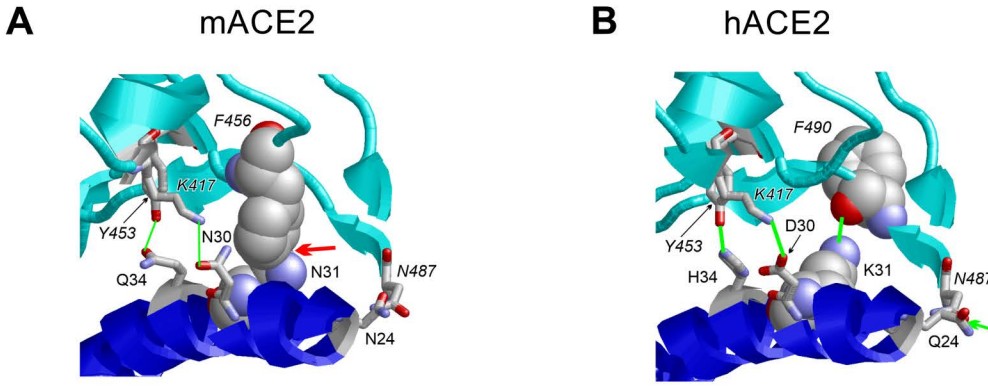

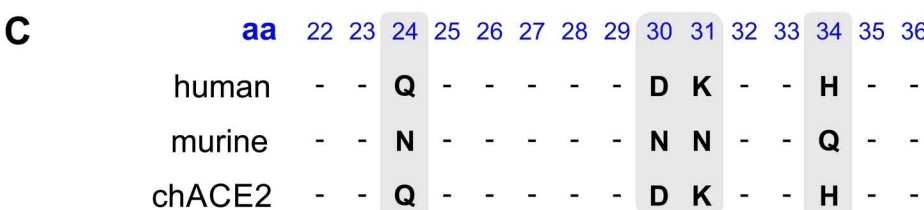

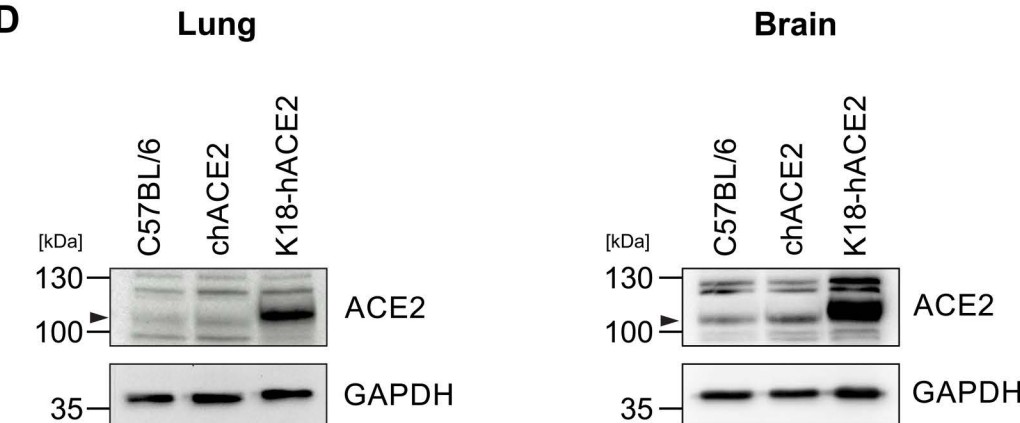

**Fig 2. Generation of a novel chimeric ACE2 mouse model by forward directed mutagenesis.** Structural analysis of the interactions formed between the SARS-CoV-2 RBD and **(A)** murine **(m)** ACE2 or **(B)** human **(h)** ACE2. The SARS-CoV-2 RBD and ACE2 are shown in cyan and blue, respectively. The four residues, which differ between hACE2 and mACE2, as well as their spatially adjacent residues within the RBDs are explicitly labeled. In hACE2, the four residues form strong polar intermolecular interactions (thick green lines). In mACE2 these polar interactions are either weaker (thin green lines) or completely absent. In addition, an unfavorable steric clash is observed between N31 and F456 of RBD and mACE2 (red arrow). **(C)** Alignment of the human, murine and chimeric ACE2 amino acid sequences, created with the UniProt Align tool. Blue numbers indicate the amino acid position within mACE2. Grey boxes highlight the amino acids, which have been changed to create a human/mouse chimeric receptor (chACE2). **(D)** Immunoblot analysis of ACE2 expression in lung and brain tissue of eight weeks old chACE2, K18-ACE2, and C57BL/6 mice.

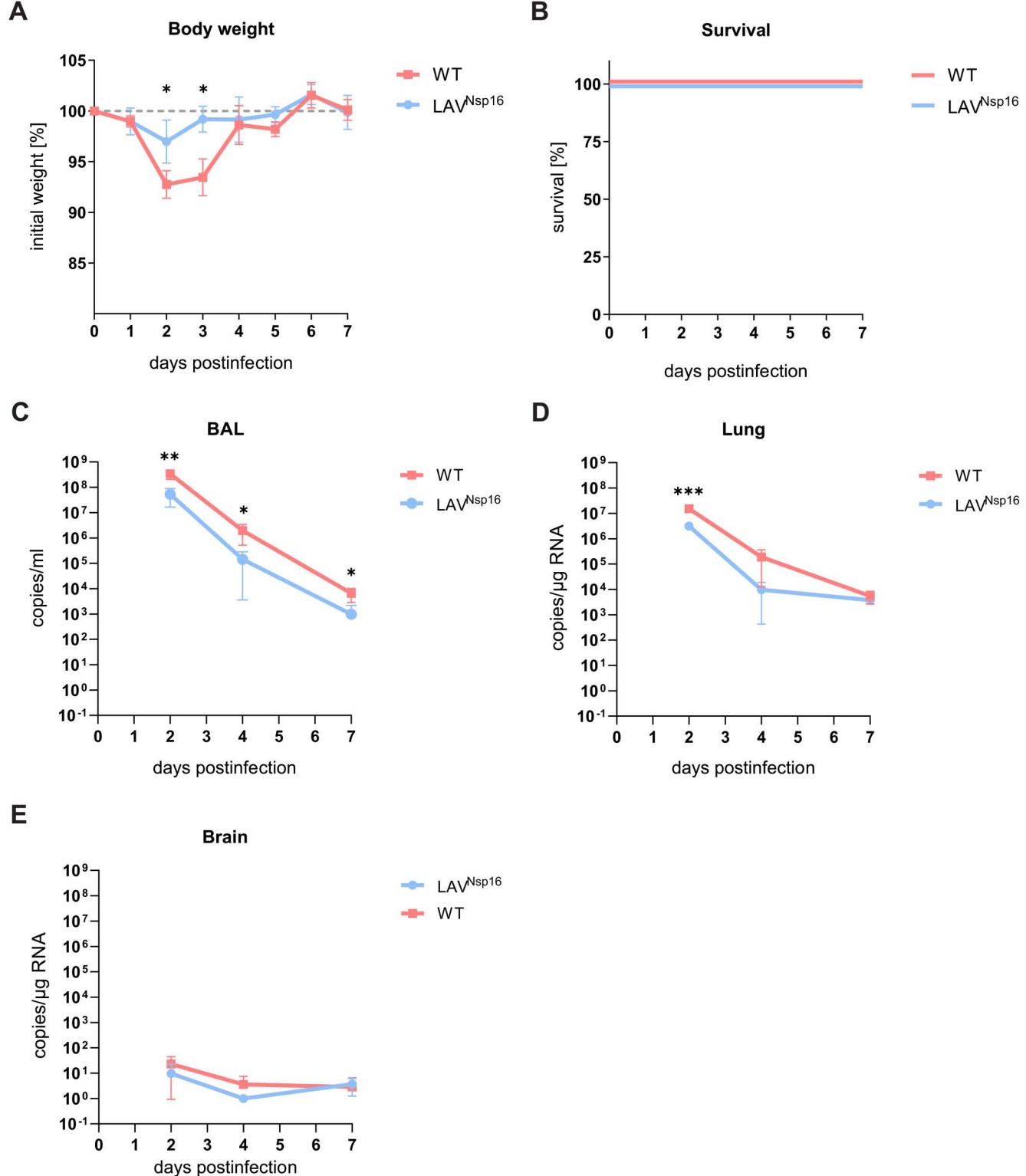

**Fig 3. Chimeric ACE2 supports SARS-CoV-2 wt and LAV^Nsp16 replication in vivo.** Eight weeks old chACE2 mice were infected intranasally with $7 \times 10^4$ PFU of wt or LAV^Nsp16 virus in 30 µl PBS. Clinical scores and bodyweight of infected mice were monitored daily. Animals were sacrificed on days two, four, and seven postinfection (n = 4). Bodyweight **(A)** and survival status **(B)** of animals sacrificed on day seven postinfection are depicted. Weight

measurements of individual animals were normalized to the initial weight and are shown as mean +/- s.d. At each endpoint, bronchoalveolar lavage (BAL) **(C)**, lung **(D), and brain (E)** samples were collected and viral RNA copy numbers were determined by RT-qPCR. Bars represent the mean +/- s.d. overlaid with individual data points. Statistical analysis of bodyweight and viral genome titers was performed by conducting separate two-tailed t-tests for each timepoint.

mice (Fig 3A, 3B). We found the pathogenicity of LAV[Nsp16] to be even further attenuated in chACE2 mice and that infected animals barely display any clinical symptoms such as weight loss. In chACE2 mice, wt virus and LAV[Nsp16] RNA titers strongly declined over the course of seven days postinfection. The attenuated LAV[Nsp16] was replicating to lower titers and was constantly found to be 10-fold less abundant than wt virus (Fig 3C, 3D). At four days postinfection, we sequenced wt virus isolated from the lung of chACE2 and K18-hACE2 mice but did not find any adaption to the murine receptor in the RBD of Spike, as it has been observed for mouse-adapted strains. This suggests that viral replication solely depends on the chimeric ACE2 receptor (S2 Fig). [17] Importantly, we could not detect significant levels of viral genomic RNA or *Ifnb1* transcripts, as signs of inflammation, in the brain of neither wt virus nor LAV[Nsp16]-infected mice (Fig 3E and S1 Fig), which is in stark contrast to K18-hACE2 mice (Fig 1C). Together, these results confirm our previous in vitro data and K18-hACE2 infection assays, making LAV[Nsp16] a promising candidate for LAV development and chACE2 mice a perfect model to analyze replication and immune responses upon LAV[Nsp16] immunization in great detail.

## LAV[Nsp16] immunization generates local innate responses in chACE2 mice

Next, we compared local innate immune responses to LAV[Nsp16] and wt virus infection in chACE2 mice and analyzed BAL and lung tissue at various time points postinfection (Fig 4 and S4-S7 Figs). Two days upon wt virus infection, we found a strong and significant increase in macrophage, monocyte, neutrophil, and NK cell numbers in the BAL by flow cytometry (Fig 4A-D), demonstrating a robust local innate response to infection in chACE2 mice. A similar immune induction was observed upon infection with LAV[Nsp16], albeit to a lesser extent, which reflects the attenuated replication of LAV[Nsp16] (Fig 3C, 3D). In addition, we observed significant numbers of CD8+T cells present in the BAL of wt- and LAV[Nsp16]-infected mice, which increased over time (Fig 4E). To confirm the innate immune activation upon LAV[Nsp16] and wt virus infection, we also quantified the transcript levels of Interferon β1 (*Ifnb1)* and the interferon-stimulated genes (ISGs) *Oas1* and *Ifi44* in lung tissue lysates by RT-qPCR as surrogate for immune activation (Fig 4F). Similarly, we found a strong and early upregulation of ISG transcripts in response to wt virus infection and a slightly attenuated upregulation of *Ifnb1* and the ISG transcripts upon LAV[Nsp16] infection, suggesting a strong interferon-mediated innate response in chACE2 mice.

## LAV[Nsp16] immunization induces local and systemic adaptive immune responses

In contrast to the very high morbidity in K18-ACE2 mice, SARS-CoV-2 is less pathogenic in young chACE2 mice with all animals recovering from infection over time (Fig 3A, 3B). Thus, our mouse model allows for detailed comparison of adaptive immunity in responses to wt virus and LAV[Nsp16] infection weeks after infection. Since anti-S IgG antibody levels induced by vaccination or infection are considered a correlate of protection, we first asked whether infections with LAV[Nsp16] or wt virus induces robust and comparable anti-S IgG antibody levels in chACE2 mice. [29] At 21 days postinfection, we analyzed BAL and serum of eight weeks old mice in an ELISA-based antibody binding assay. [30] Infection with wt virus resulted in a robust systemic (serum) and local (BAL) induction of anti-S IgG in our chACE2 mice (Fig 5A, 5B). The increase in S-specific IgG levels also correlated with the detection of S-specific B cell responses in these mice at 21 days postinfection (Fig 5C). Importantly, infection with LAV[Nsp16] also induced S-specific B cell responses and anti-S IgG levels in sera of infected mice comparable to wt virus infection (Fig 5A, 5C). The induction of IgG in the BAL of LAV[Nsp16]-infected mice was less pronounced, which might be due to the attenuated nature of the LAV[Nsp16] lacking Nsp16 activity (Fig 5B). Similarly, we found the neutralization capacity of serum from LAV[Nsp16]-infected animals to be slightly reduced compared to serum from wt virus-infected mice (Fig 5D). In contrast, both infections resulted in a strong and comparable induction

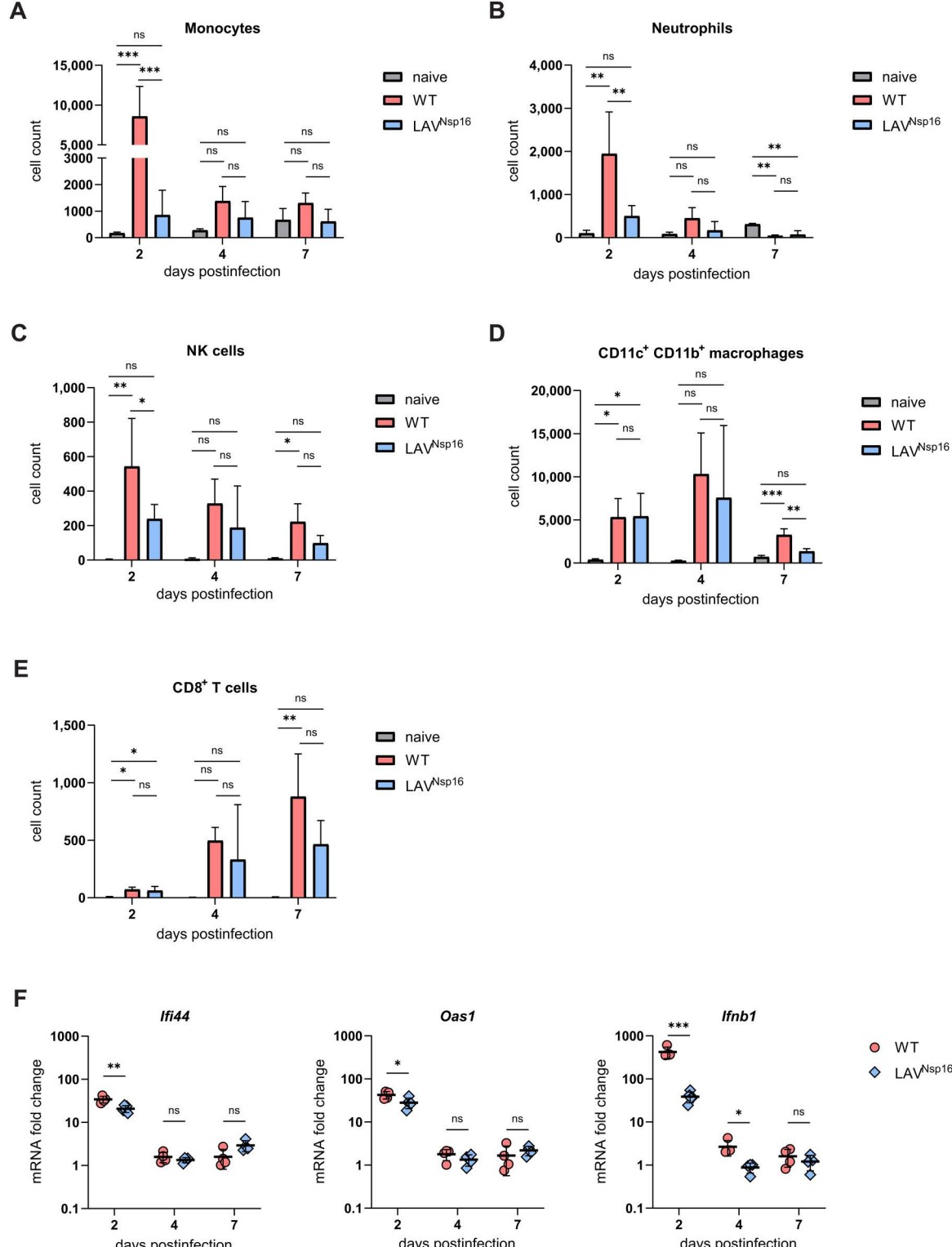

**Fig 4. Lung infiltration and expression of interferon stimulated genes upon wt virus and LAV[Nsp16] infection in chimeric ACE2 mice. (A-E)** Lung infiltration by monocytes (CD45+CD11b+CD11c-), neutrophils (CD45+Gr-1[high]), natural killer cells (CD45+CD49b+), macrophages (CD45+CD11b+CD11c+), and CD8+T cells (CD45+CD8+) was determined by antibody staining of cells present in bronchoalveolar lavage (BAL) and subsequent flow cytometry. Bars represent the mean +/- s.d. Statistical analysis were preformed using one-way ANOVA followed by Tukey's test (A, B, D) or Kruskal-Wallis tests

followed by Dunn's correction (C). **(F)** Transcript levels of the indicated interferon-stimulated genes in lung tissue of infected animals were determined by RT-qPCR and normalized to GAPDH (ΔΔCt). Relative mRNA levels compared to uninfected controls are plotted. Statistical analysis was performed using unpaired two-tailed t-tests. Samples originate from infected animals analyzed in Fig 3 at two, four, or seven days postinfection.

of antigen-specific CD8 + T cell responses (Fig 5E-H). Upon infection with both viruses, we found high levels of KLRG1$^{-}$/ CD127$^{+}$ central memory cells (T$_{CM}$) in the spleen of infected animals. In lung tissue, we found a strong induction of T$_{CM}$ as wells as increased numbers of CD69$^{+}$/ CD103$^{+}$ tissue-resident memory cell (T$_{RM}$) upon infection with either virus. In contrast, T effector cells (T$_{EFF}$) as well as effector memory cells (T$_{EM}$) were only found in very low numbers. Upon restimulation of CD8$^{+}$ T cells from the spleen and lung with S peptides, we found a similar percentage of activated, TNF$a$, IFNγ, or IL2-expressing CD8$^{+}$ T cells in wt and LAV$^{Nsp16}$-infected animals, indicating a comparable cellular antiviral response (Fig 5I, 5K).

## Strongly enhanced SARS-CoV-2 pathogenicity in aged chACE2 mice

Older individuals are at higher risk of SARS-CoV-2 infection and are more likely to experience severe outcomes. [12–14] Similarly, previous studies using K18-hACE2 or C57BL/6 mouse models found high-titer viral replication along with a higher pathogenicity in aged animals. [17–19, 31] To test how age affects SARS-CoV-2 infection in chACE2 mice, we infected aged mice (46–55 weeks) with either wt virus or LAV$^{Nsp16}$ and analyzed pathogenicity at various time points postinfection (Fig 6A, 6B). In contrast to young animals (Fig 6E, 6F), SARS-CoV-2 wt infection was highly pathogenic in aged chACE2 mice. All animals were progressing to severe disease and were reaching the predefined welfare endpoints within four days postinfection. LAV$^{Nsp16}$-infected animals, however, showed no, or only very mild, signs of infection, demonstrating a significant attenuation of LAV$^{Nsp16}$, also in more vulnerable aged animals (Fig 6A, 6B). Fittingly, when analyzing infectious viral titers as well as viral RNA titers in BAL and lung tissue of aged animals, LAV$^{Nsp16}$ was replicating to lower titers than wt virus (Fig 6C, 6D, S3 Fig). We also compared viral genome titers in the BAL of young and aged mice and found slightly enhanced viral replication in old animals (Fig 6G). One reason for the enhanced viral replication might lie in age-dependent alterations in expression of the ACE2 receptor, which has been reported previously. [32] Thus, we compared the expression levels of chACE2 and mACE2 in the lungs of young and aged chACE2 and C57BL/6 mice by immunoblot (Fig 6H). For both mouse strains, we found a stronger expression of the ACE2 receptor in lung tissue of aged animals compared to younger animals (Fig 6H). Importantly, chACE2 expression level and band size were very similar to mACE2 in young and old mice, further confirming the physiological expression of chACE2. Since advanced age can affect host immune responses to SARS-CoV-2, we analyzed the innate immune response in aged mice upon wt and LAV$^{Nsp16}$ infection. Even more pronounced than in young mice, we found a strong infiltration of the lung by macrophages, monocytes, neutrophils, and NK cells upon infection with wt virus (Fig 7A-D). We also found substantial numbers of CD8 + T cells present in the BAL of wt- and LAV$^{Nsp16}$-infected mice, which increased over time (Fig 7E). In contrast, and in line with a less efficient replication (Fig 6), a slightly decreased infiltration of most cell types was observed in aged animals treated with LAV$^{Nsp16}$ (S7A-E Fig). On a cellular level, however, LAV$^{Nsp16}$ infection induced similar ISG expression levels as wt virus, despite its attenuated replication (Fig 7F). To compare lung pathology upon wt and LAV$^{Nsp16}$ infection, we performed histological assays (Fig 8A). Hematoxylin & eosin (H&E) staining assessing parenchymal, peribronchial, perivascular, and intrabronchial inflammatory cell infiltration or structural damage, showed comparable inflammation upon wt and LAV$^{Nsp16}$ infection (Fig 8B). This finding is in line with the overall comparable innate immune cell infiltration seen in the BAL of wt and LAV-infected mice, especially in aged animals (Fig 6). In young mice, we observed a rather mild pathogenicity of both viruses, with isolated foci of inflammation resulting mainly in peribronchial, perivascular, or intrabronchial immune cell accumulation. Although overall inflammation levels were similar, immune cell infiltration into the alveolar parenchyma became only visible in the lungs of aged mice and could not be detect in young mice (Fig 8A). To test whether the

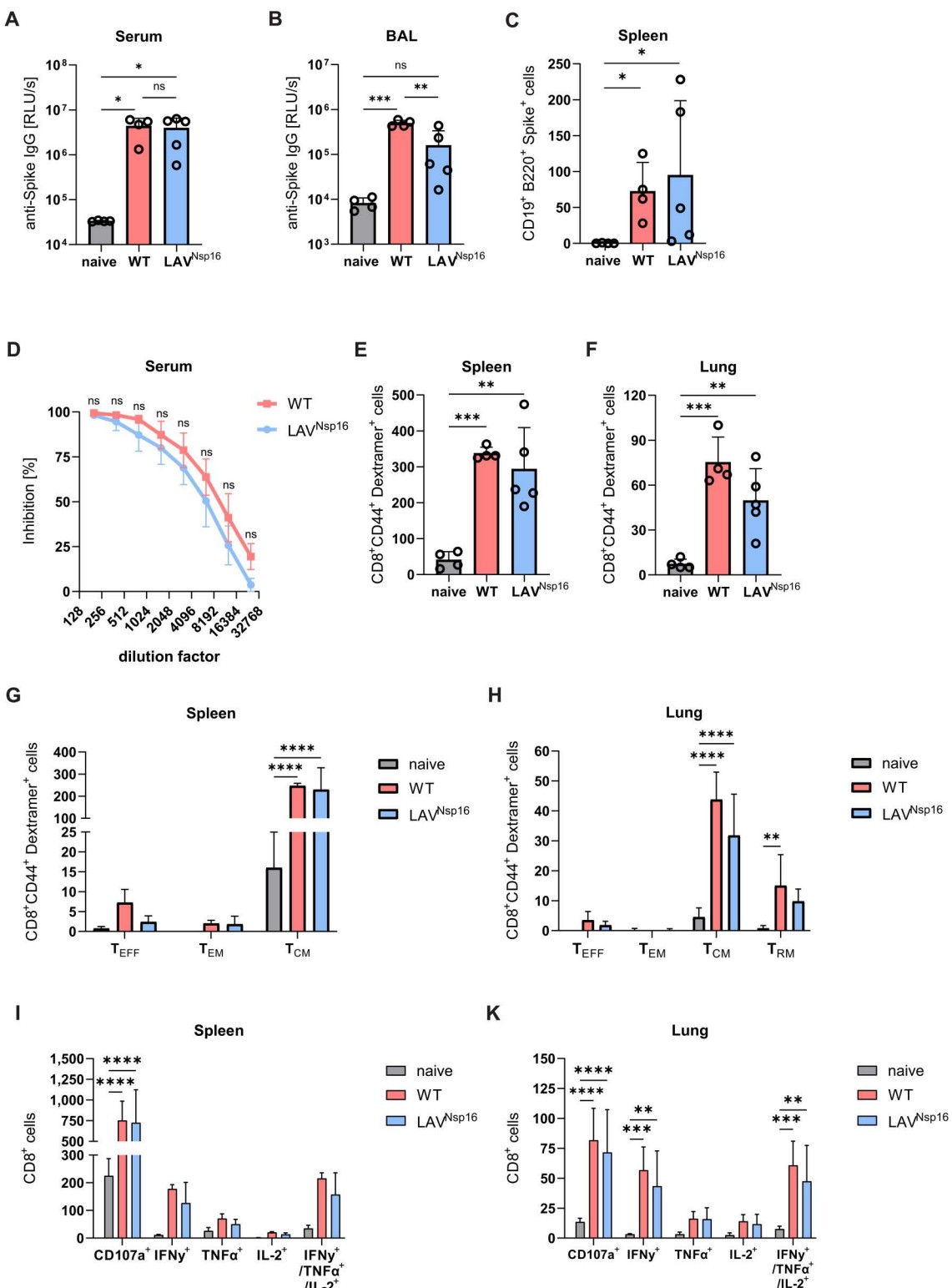

**Fig 5. LAV^Nsp16 immunization induces local and systemic adaptive immune responses.** Eight weeks old chACE2 mice were infected with 7x10^4 PFU of wt virus or LAV^Nsp16. Bronchoalveolar lavage (BAL), serum, and lymphocytes derived from lung and spleen tissue were collected on day 21 postinfection. Spike-specific antibody levels in serum **(A)** and BAL fluid **(B)** were determined by ELISA. Mean luminescence +/- s.d upon HRP-coupled secondary antibody incubation is shown overlayed by data points of individual animals. S-specific B cells **(C)** and T cells **(E, F)** present in spleen or lung

tissue were identified by antibody staining and subsequent flow cytometric analysis. Cell counts are displayed as mean +/- s.d. overlayed by individual data points. Statistical analysis was performed using one-way ANOVA followed by Dunnett's test (B, D, E) or Kruskal-Wallis analysis followed by Dunn's test **(A, C)**. **(D)** Neutralization capacity of S-specific antibodies from sera of wt- or LAV^Nsp16-infected animals was analyzed in pseudotyped virus neutralization assays. Neutralization capacity is shown as mean inhibition of quadruplicate infections +/- s.d. **(G, H)** Spike-specific CD8+ T cells were further discriminated into KLRG1+ CD127- effector T cells ($T_{EFF}$), KLRG1+ CD127+ effector memory T cells ($T_{EM}$), KLRG1- CD127+ central memory T cells ($T_{CM}$), and single or double positive CD69+ CD103+ tissue resident T cells ($T_{RM}$). **(I, K)** Lung and spleen-derived T cells were restimulated ex vivo with S-derived peptides (VNFNFNGL and VTWFHAIHVSGTNGT). Resulting production of CD107a, IFNγ, TNFα, and IL2 by CD8+ T cells was determined by intracellular cytokine staining and analyzed by flow cytometry. Results are plotted as mean counts/million +/- s.d. Statistical analysis was done by two-way ANOVA followed by Bonferroni correction.

epithelial barrier function is compromised upon infection, we also quantified protein level in the BAL of aged mice as a surrogate marker of tissue damage at two and four days postinfection (Fig 8C). We observed a strong increase in protein level in the BAL by day four upon wt infection. In contrast, LAV^Nsp16-infected aged mice show less leakage of serum proteins into the BAL. This suggests less damage to the alveolocapillary membrane despite comparable inflammation and confirms the strong attenuation of LAV^Nsp16 infection in aged mice.

### LAV^Nsp16 immunization blocks viral replication upon homologous or heterologous SARS-CoV-2 challenge

Finally, we asked whether vaccination with LAV^Nsp16 protects aged mice from subsequent SARS-CoV-2 challenge. We therefore immunized aged mice with LAV^Nsp16 intranasally at 21 days prior to challenge with either a homologous wt (B.1) or a heterologous (Delta) SARS-CoV-2 variant. Compared to naïve control animals, immunization with LAV^Nsp16 protected aged mice from high pathogenicity of wt infection (Fig 9A, 9C). While 50% of the PBS-treated, unvaccinated animals succumbed, all LAV immunized animals showed a strongly reduced pathogenicity. Heterologous infection with the Delta variant did not result in diseases within four days, suggesting low pathogenicity or slow replication kinetics of the Delta virus in chACE2 mice (Fig 9B, 9D). However, when comparing viral load in the lung or BAL of infected mice at four days postinfection, it became clear that immunization with LAV^Nsp16 blocked the replication of homologous wt virus as well as heterologous Delta virus in aged animals (Fig 9E, 9F). Of note, we detected low level of viral nucleic acids in all immunized mice, including non-infected mice, suggesting the presence of background levels of LAV^Nsp16 nucleic acids even at 25 days postinfection. These results clearly show that LAV^Nsp16 immunization potently protects from heterologous and homologous SARS-CoV-2 replication and from severe age-dependent pathogenicity of SARS-CoV-2 in our new chACE2 mouse model.

### Discussion

Here, we analyzed the efficacy of a live attenuated vaccine candidate against homologous and heterologous SARS-CoV-2 infection using a newly developed murine infection model. We therefore generated chACE2 mice that express a human-murine chimera of the SARS-CoV-2 receptor ACE2, harboring four distinct, human-specific amino acid exchanges at the S protein binding site in murine ACE2 (Fig 2A-C). Although, similar models have been developed recently, the rational design of chACE2 to enhance RBD binding to murine ACE2 resulted in the exchange of a unique set of four amino acids, which are not addressed in other models. [33–36] The most frequently used K18-hACE2 transgenic mouse model expresses high levels of human ACE2 under control of the K18 promoter in epithelial cells. [21] However, in addition to other organs, detectable levels of hACE2 transcript are also found in brain tissue. [21] The atypical expression of hACE2 has been suggested to be, at least in part, responsible for the fatal neurodissemination of SARS-CoV-2 in K18-hACE2 mice upon intranasal infection. [27, 33] In contrast, chACE2 mice express the receptor under the control of the endogenous murine promoter and do not show strong overexpression of the receptor in brain tissue (Fig 2D). The natural ACE2 expression level more accurately reflects expression in humans. It results in mild disease in younger animals and allows for studying SARS-CoV-2 replication and pathogenicity as well as immune responses upon infection over a longer period

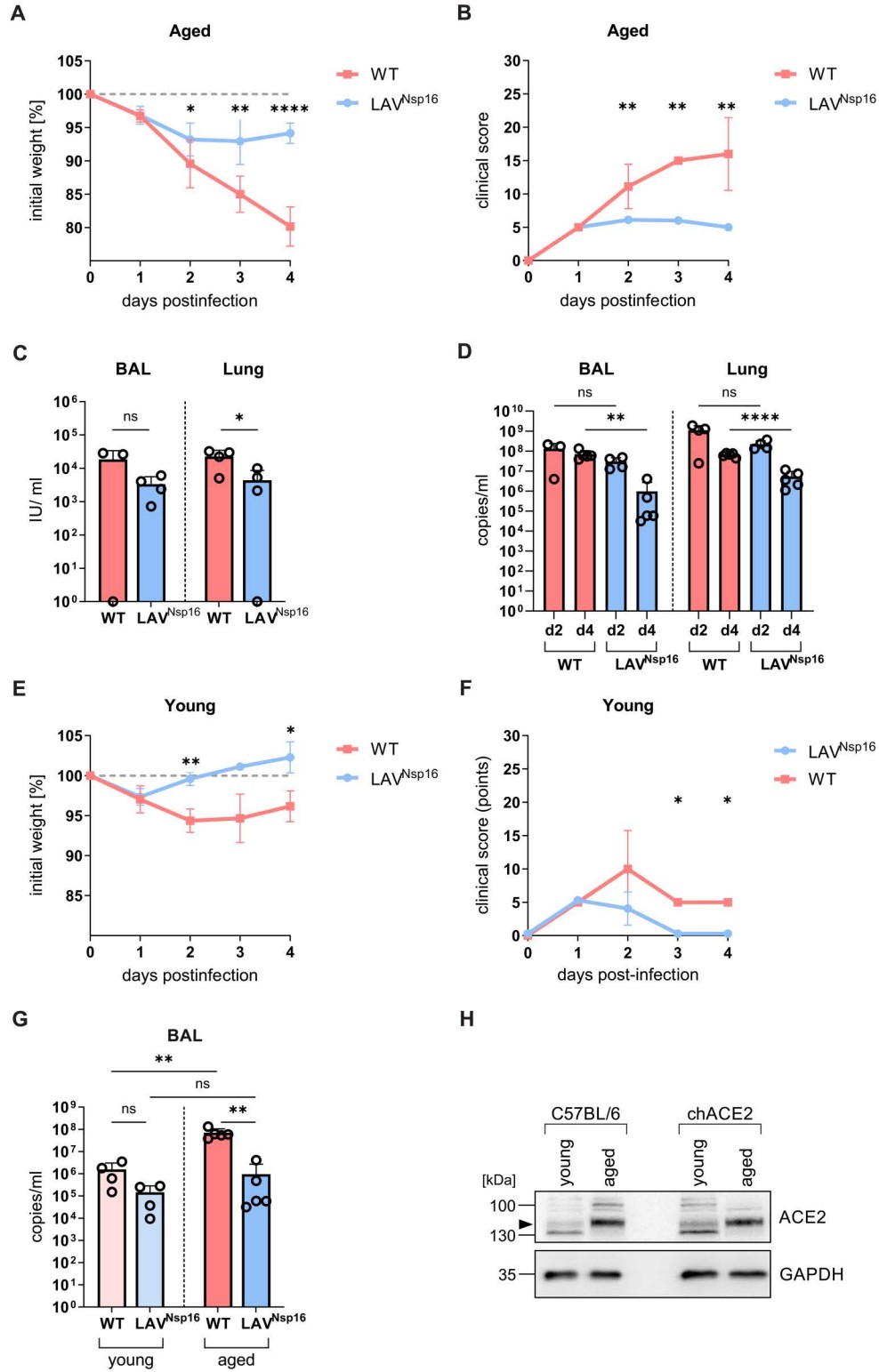

**Fig 6. SARS-CoV-2 infection results in high viral loads and severe disease in aged chACE2 mice.** Forty-six to fifty-five weeks old chACE2 mice were infected with $7 \times 10^4$ PFU of wt virus or LAV$^{Nsp16}$ in 30 µl PBS (n = 4). Mice were sacrificed on day 2 (n = 4) or day 4 (n = 5) postinfection. Weight loss **(A)** and clinical score **(B)** of all animals were determined over time and shown as mean +/- s.d.. **C)** Infectious viral titers in BAL and lung tissue of aged

mice at day 2 postinfection were assessed by infection of Caco-2 cells followed by immunofluorescence analysis. Titers are shown as mean of triplicate infections +/- s.d. overlayed with data points of individual animals. **(D)** Viral genome copies BAL and lung tissue was determined by RT-qPCR and is shown as mean of quadruplicates +/- s.d. overlayed with data points of individual animals. **(E, F, G)** For clarity reasons, weight, clinical, and viral titers from young chACE2 mice (as shown in Fig 3) were included. Viral load in BAL (G) of young and aged chACE2 mice two days postinfection with wt and LAV[Nsp16] was determined by RT-qPCR and is shown as mean +/- s.d. overlayed with data points of individual animals. **(A-G)** Statistical analysis was performed by unpaired two-tailed t-tests. **(H)** ACE2 and GAPDH protein expression in lung tissue lysates of uninfected, young (8 weeks), or old (42 weeks) chACE2 and C57BL/6 (B6) mice is shown.

of time (Fig 5). The fact that expressing physiological levels of the receptor in chACE2 mice results in rather mild disease in young animals has been confirmed recently in two other murine infection models replacing either the entire murine genomic locus (ACE2-GR) or exon 2 and 3 (hyACE2) with the human ACE2 receptor sequence. [33, 37] Of note, chimeric ACE2 differs from the murine receptor in only four amino acids to minimize the possibility of unwanted side effects affecting the natural functions of murine ACE2. In contrast to the mouse-adapted virus infection model, the chimeric receptor supports the replication of non-adapted SARS-CoV-2 variants and allows the comparison of genetically-defined recombinant viruses in chACE2. Here, we used our advanced infection model to analyze viral infection and antiviral immune responses in elderly mice and to test whether an LAV approach might be able to protect those mice from homologous and heterologous challenge.

LAV designs are promising vaccines that might overcome the limitations of S-encoding mRNA vaccines, which must be constantly adapted due to newly emerging viral variants. LAV designs might circumvent the problem of adaptation by inducing a broader immune response to several viral antigens, while reducing the risk of adverse effects induced by natural infection. Indeed, we found that the SARS-CoV-2 LAV candidate lacking a functional 2'-O-methyltransferase to be immunogenic in mice and to protect from infection with a homologous as well as a heterologous virus variant, thereby clearly demonstrating the potential of the LAV strategy (Fig 9). In line with our findings, Nsp16 has recently been described as promising target for modification in LAV design by another group, further validating the results of our study. [10, 11] In contrast to Ye and colleagues, however, we found that not all K18-hACE2 mice, which unphysiologically overexpress hACE2, survived the challenge with SARS-CoV-2 lacking Nsp16 activity (Fig 1E). This coincided with high levels of wt virus as well as LAV[Nsp16] genomic RNA in brain tissue of these mice, rendering the K18-hACE2 model insufficient for further analysis (Fig 1C). However, using our novel mouse model, we were able to compare adaptive immune responses upon wt virus and LAV[Nsp16] infection. Our results show that the attenuated virus LAV[Nsp16] induces detectable local (lung) and systemic (serum) antibody and T cells responses similar to wt infection (Fig 5). This represents a clear advantage over intramuscular mRNA vaccinations, which do not induce efficient local immune responses. [38] A disadvantage of LAV approaches is pre-existing immunity from previous infections or vaccinations that might limit its efficacy. In future studies, we will therefore test the effect of pre-existing immunity on LAV[Nsp16] efficacy and will generate LAVs harboring alternative S sequences to optimize the promising candidate LAV[Nsp16].

In chACE2 mice, SARS-CoV-2 wt (B.1) infection is less pathogenic compared to K18-hACE2 mice and all chACE2 mice recovered from infection within seven days (Fig 3A-D). The mild phenotype correlates with the absence of SARS-CoV-2 in the brain of chACE2 mice (Fig 3E and S1 Fig), again in stark contrast to the K18-hACE2 overexpression model (Fig 1C). Thus, SARS-CoV-2 pathology in chACE2 mice represents the situation in humans more accurately. Taking advantage of this finding, we also analyzed SARS-CoV-2 infection of elderly mice, since old age in humans correlates with a more severe disease outcome. Interestingly, SARS-CoV-2 wt replicates to higher titers, is highly pathogenic, and shows enhanced mortality in aged but not in young chACE2 mice (Fig 6). Similar age-dependent phenotypes have been observed in K18-hACE2 mice and with mouse-adapted infection models before, further validating our model. [17–19, 31] The age-dependent phenotype in chACE2 mice is more pronounced compared to K18-hACE2 mice, mainly due to a more benign infection in younger mice. Previously, a lack of coordinated early innate immune responses was discussed as reason for the enhanced pathogenicity in aged animals. [17] In chACE2 mice, however, we did not observe major differences

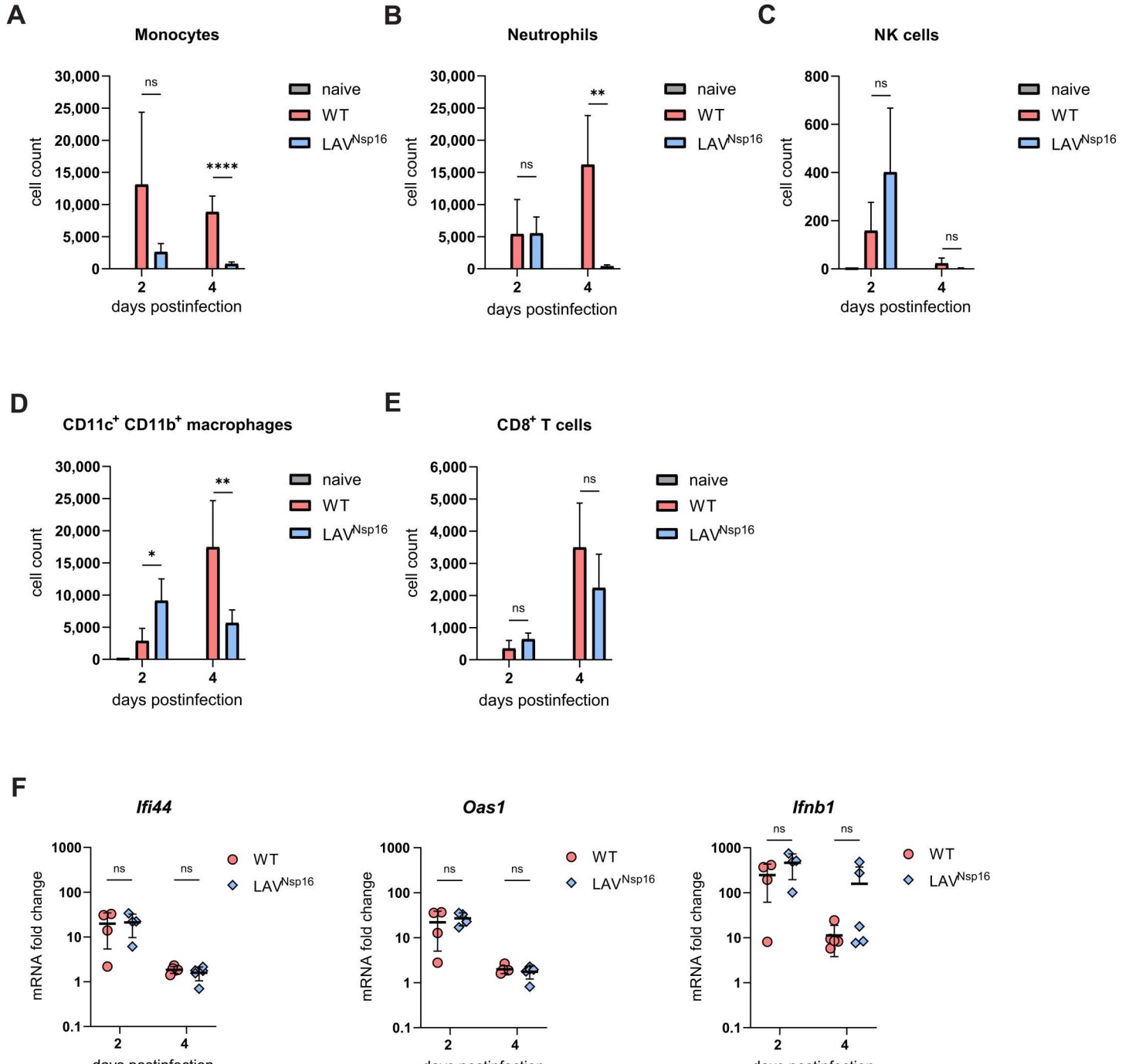

**Fig 7. Strong immune cell infiltration, enhanced ISG expression, and enhanced tissue damage in the lung of aged chACE2 mice.** Immune responses were analyzed in 46 to 55 weeks old chACE2 mice at the indicated time points postinfection with $7\times10^4$ PFU of wt virus or LAV$^{Nsp16}$ (as shown in Fig 6). **(A-E)** Lung infiltration by CD45$^+$CD11b$^+$ monocytes, CD45$^+$Gr-1$^{high}$ neutrophils, CD45$^+$CD49b$^+$ natural killer cells, CD45$^+$CD11b$^+$CD11c$^+$ macrophages, and CD45$^+$CD8$^+$ T cells was determined by antibody staining of cells in BAL followed by flow cytometry. The mean cell number +/- s.d. as well as individual data points are shown. Statistical analysis was done using unpaired two-tailed t-tests **(F)** mRNA levels of *Ifi44, Oas1* and *Ifnb1* present in lung tissue of infected animals were determined by RT-qPCR and normalized to GAPDH (ΔΔCt). For statistical analysis, unpaired two-tailed t-tests were performed.

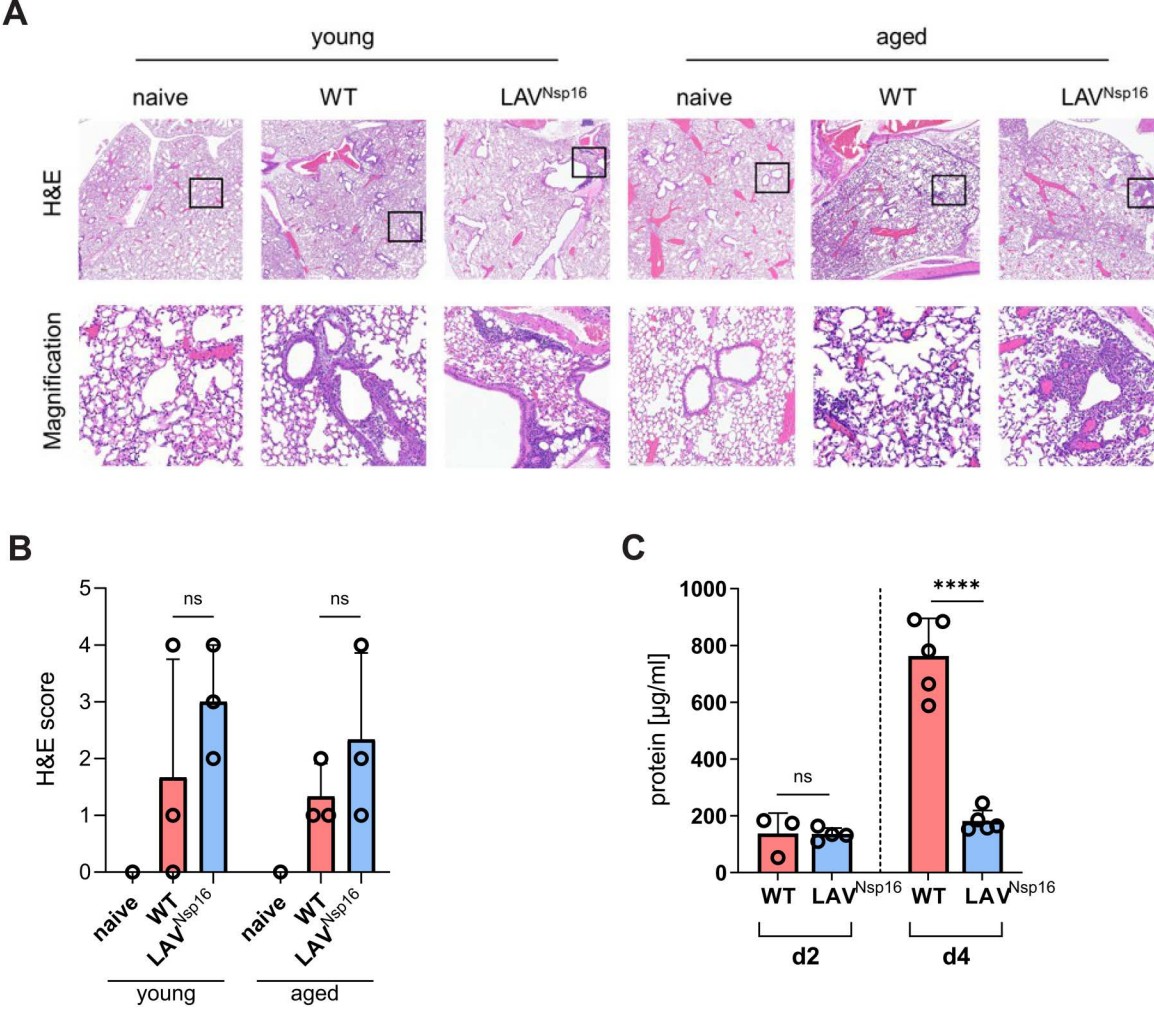

**Fig 8. Lung pathology upon infection. (A)** Lungs of young and aged chACE2 mice infected with wt virus or LAV^Nsp16 were resected at four days postinfection (n = 3). Upon fixation with paraformaldehyde (PFA), tissues were embedded in paraffin. To visualize lung infiltration, 5 µm tissue sections were stained with hematoxylin and eosin (H&E). Slides were scanned by S210 digital slide scanner from Hamamatsu (0.23 µm/pixel; S210, Hamamatsu, Japan). Pictures were generated using the Case Viewer software (3DHistech, Hungary). One representative slide per condition is shown. **(B)** H&E score of lungs of naïve, WT-, and LAV^Nsp16 -infected mice as shown in (A). Histopathology was assed by scoring perivascular, peribronchial, intrabronchial, or parenchymal inflammatory events (total score 0 -16). Data are presented as mean ± s.d. (n = 3). **(C)** As surrogate for epithelial barrier damage, protein concentration in BAL samples of aged animals sacrificed on day two and day four postinfection was determined by BCA assay. Results are shown as mean +/- s.d.

in early innate responses between young and aged mice (Figs 4 and 7). Another possible explanation might lay in the stronger expression of the viral receptor in aged animals (Fig 6H). In addition, we observe a shift in size of murine ACE2 in aged animals in the immunoblot analysis (Fig 6H). Similar to the human protein, two different isoforms of ACE2 are expressed in mice and are also posttranslationally modified, for example by glycosylation. [39, 40] Thus, it is conceivable that either a change in expression or in posttranslational modification of chACE2 in aged animals might contribute to enhanced pathogenicity. Histological assays show that the overall inflammatory events in the lung of young and aged mice were similar in number and locally confined. However, we found an immune cell influx into alveolar parenchyma only in aged but not in young animals, in which mainly perivascular and peribronchial immune cell accumulations were

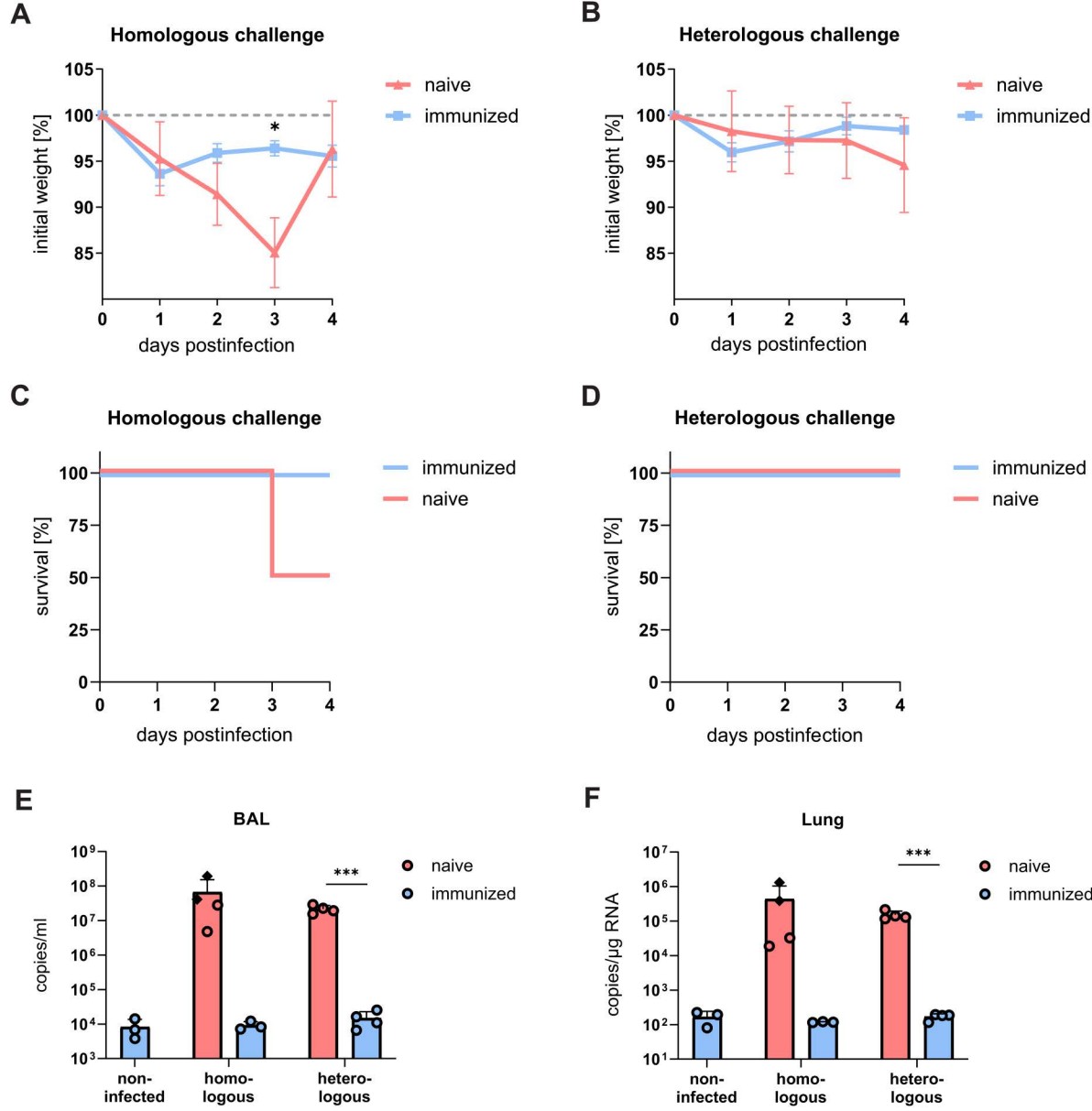

**Fig 9. LAV^Nsp16 immunized aged animals are protected from homologous or heterologous SARS-CoV-2 challenge.** Aged chACE2 mice were immunized with 7x10⁴ PFU of LAV^Nsp16. At 21 days postinfection, immunized and naïve mice were challenged with 7x10⁴ PFU virus of either the B.1 (WT, homologous) or the Delta (heterologous) variant. Mice were sacrificed at four days postchallenge, or when predefined welfare endpoints were reached (clinical score ≥ 20) at day 3 (♦). Weight measurements, normalized to the initial bodyweight of wt virus **(A)** or Delta virus **(B)** infected animals are shown as mean +/- s.d. Survival rates of animals challenged with wt **(C)** or Delta **(D)** virus are shown. Viral loads in bronchoalveolar lavage (BAL) **(E)** and lung tissue **(F)** were determined by RT-qPCR and plotted as mean +/- s.d. overlayed by individual data points. Statistical analysis was performed by two-way ANOVA followed by Bonferroni's correction for multiple comparison.

observed. Thus, it will be interesting in the future to study the localization and modification of the chimeric/murine ACE2 receptor in the lung of aged mice and test whether this can be linked to enhanced pathogenicity. Importantly, LAV^Nsp16 is strongly attenuated in elderly animals, when compared to wt infection. It is replicating to lower titers, induces less tissue

damage compared to wt virus as assessed by protein leakage in the BAL, and infected animals recover more quickly, demonstrating the low pathogenicity of attenuated LAV[Nsp16] in aged mice. Vaccination with the B.1 variant-based LAV[Nsp16] induces immune responses in those animals and protects elderly chACE2 animals form the severe pathology of a SARS-CoV-2 B.1 challenge 21 days later (Fig 9). Importantly, LAV[Nsp16] also protects mice from infection with the heterologous variants such as SARS-CoV-2 Delta (Fig 9), thereby displaying the advantage of LAV approaches. Of note, we found that even high dose SARS-CoV-2 Delta infection (7 x 10$^4$ IU) results in reduced pathogenicity compared to B.1 infection in aged chACE2 mice, despite comparable viral load in BAL and lung. This finding is especially puzzling since Delta virus variants show enhanced pathogenicity in humans. However, similar results for Delta virus infection in mice have been reported previously using K18-hACE2 mice. [41, 42]

The newly developed murine SARS-CoV-2 infection model, chACE2, expresses physiological levels of a chimeric ACE2 receptor. It efficiently supports SARS-CoV-2 replication but does not display enhanced mortality due to viral neurodissemination. As for SARS-CoV-2 infection in elderly patients, aged chACE2 mice suffer from aggravated disease. Interestingly, immunization with the live attenuated vaccine candidate LAV[Nsp16] induces robust immune responses and protects aged chACE2 mice from otherwise severe pathogenicity. This makes LAVs lacking an RNA 2'-O-methyltrasferse activity promising candidates for live vaccine development. Especially in combination with additional attenuating mutations, which further enhance their safety profile, such LAVs will be promising designs not only against SARS-CoV-2 and new Variants of Concern but also against future emerging viruses encoding similar enzymatic activities. In addition, the novel chACE2 mouse model will allow answering current pressing questions in SARS-CoV biology, such as age-related pathogenicity or the development of novel vaccine candidates, as well as addressing future threats, making it an important tool in terms of pandemic preparedness.

## Materials and methods

### Generation of chimeric ACE2 mice

Mice expressing a human-murine chimera of ACE2 (C57BL/6-Ace2-tm1(ACE2*1)-TW) were generated by CRISPR/Cas9-mediated gene editing in mouse zygotes. Pronuclear stage zygotes were harvested from super-ovulated C57BL/6N females mated with C57BL/6N males. The pronuclei of the zygotes were microinjected with a mix containing 40 ng/µl recombinant Cas9 (Integrated DNA Technologies, Inc.), 0.33 pmol/µl Alt-R® CRISPR-Cas9 tracrRNA duplex (protospacer AAAGTTGTTTAAAAATGTCT, Integrated DNA Technologies) and 60 ng/µl single stranded DNA (483 bp comprising 8 nucleotide mutations resulting in Q24N, D30N, K31N, and H34Q exchanges). After microinjection into the pronucleus, cells were cultured overnight and two-cell-stage embryos were transferred into pseudopregnant CD-1 foster mice. Genotyping was performed by direct Sanger-sequencing of a 218 bp PCR product covering the mutations and comparing the reads to the murine *Ace2* sequence. Animal experiments were approved by *Regierung von Mittelfranke*n (license 55.2.2-2532-2-737-23). Hemizygous male and homozygous female mice were used for infection experiments.

### Viruses

Recombinant SARS-CoV-2 wt and Nsp16-deficient (LAV[Nsp16]) viruses are both based on a passage-free SARS-CoV-2 Pangolin B.1 genome and were generated as described previously. [9, 22] Briefly, viral particles from the supernatant from Caco-2 MDA5 knockout (KO) cells were purified by filtration (0.4 µm) and ultracentrifugation through a 20% sucrose cushion, resuspended in PBS, and quantified on Caco-2 cells by TCID50 assay. [9] Similarly, a SARS-CoV-2 Delta B.1.617.2 isolate was passaged on Caco-2 cells, purified, and quantified as described above. Genomic sequences of all viruses were verified by NGS RNA sequencing.

### Infection of mice

In general, young (8–12 weeks) and aged (40–42 weeks) mice were anesthetized with ketamine/xylazine prior to intranasal infection (i.n.) with virus diluted in 30 µl PBS. K18-hACE2 mice (B6.Cg-Tg(K18-ACE2)2Prlmn/J) overexpressing

the human ACE2 receptor were infected with $1 \times 10^3$ infectious units (IU) of virus in PBS, while chACE2 mice (C57BL/6-Ace2-tm1(ACE2*1)-TW) received $7 \times 10^4$ IU. Clinical scores and bodyweight of infected animals were monitored daily. At predefined time points postinfection or when predefined welfare endpoints were reached, animals were euthanized by intraperitoneal (i.p.) administration of pentobarbital (400–800 mg/kg). Mice had ad libitum access to drinking water and food and were housed under specific pathogen-free conditions. All experiments involving animals were approved by *Regierung von Mittelfranken* (license 55.2.2-2532-2-1599) and were performed in designated BSL-3 workspaces (VII/39/ZS2001, *Stadt Erlangen*).

## RNA extraction from tissue and bronchoalveolar lavage (BAL)

Lung and brain tissue was homogenized in 1 ml lysis buffer (RA.1 supplemented with 1% ß-mercaptoethanol) at 6,500 rpm for 15 sec (Precellys 24 Touch Homogenizer, Bertin Technologies). Samples were centrifuged at 5,000 rpm for 5 minutes and 350 µl of supernatant was used for RNA extraction using the NucleoSpin® RNA Kit (Machery & Nagel). RNA concentrations were determined by Nano Drop measurement. RNA was extracted from BAL samples using the NucleoSpin RNA Virus Kit (Machery & Nagel).

## RT-qPCR analysis

Viral genome copies in tissue and BAL were quantified by RT-qPCR using the Luna Universal One-Step RT-qPCR Kit (New England Biolabs). Normalized amounts of RNA from BAL (5 µl extraction buffer) or tissue (500 ng total RNA) were analyzed with oligos targeting transcripts encoding viral RNA polymerase (RdRp-fwd 5'-GTGAAATGGTCATGTGTG GCGG-3'; RdRp-rev 5'- CAAATGTTAAAAACACTATTAGCATA-3'; RdRp-probe 5'- VIC-CAGGTGGAACCTCATCAG GAGATGC-BMN-Q535-3'). To quantify transcript levels of interferon stimulated genes (ISGs), 1.5 µg mRNA from lung tissue were reverse-transcribed into cDNA using oligo-dT primers and Superscript II RT (Life Technologies). *Ifi44, Oas1*, and *Ifnb1* level were analyzed by qPCR using ISG-specific forward and reverse oligos (IFI44-for 5'-GGCACATCTTA AAGGGCCACACTC-3', IFI44-rev 5'-CTGTCCTTCAGCAGTGGGTCATG-3'); OAS1-for 5'-CGTTGTGCCCGCCTACA GAGCC-3', OAS1-rev 5'-GCTGCAGCTCGCTGAAGGATGG-3'; IFNB1-for 5'-CTGGCTTCCATCATGAACAA-3', IFNB1-rev 5'-CATTTCCGAATGTTCGTCCT-3') and Maxima SYBR Green qPCR Master mix (Life Technologies). ISG transcription levels were normalized to GAPDH RNA levels. All reactions were run on an ABI Prism 7500 cycler (Applied Biosystems).

## Viral infectivity

Infectivity of viral isolates (IU/ml) was determined by immunofluorescence. Briefly, Caco-2 cells were infected in a 96-well format with serial dilutions of viral isolates from BAL or lung of infected animals in serum-free DMEM supplemented with non-essential amino acids. At one hour postinfection, medium was replaced and cells were cultivated for additional 11 hours in DMEM at 37°C. Next, cells were fixed in 4% PFA and permeabilized in 0.5% Triton. Upon blocking in 5% skimmed milk, cells were probed with anti-SARS-CoV-2 S antibodies TRES618 and TRES219 (250 ng/ml) and anti-mouse IgG Alexa488 (Cell Signaling 4408S). [43] Infectivity was quantified using the ImageXpress Pico Automated Cell Imaging System (Molecular Devices).

## Cell isolation and flow cytometry

Immune cells in BAL were collected at the indicated time points by centrifugation at 5,000 rpm for 5 min. CD45+ hematopoietic cells (clone 104, BioLegend), CD45+CD11c+CD11b+ macrophages (clone HL3, BD Biosciences), CD45+CD11b+ monocytes (clone M1/70, clone BM8, BioLegend), CD45+Gr-1high neutrophils (clone RB6-8C5,BioLegend) and CD45+CD49b+ natural killer cells (clone DX5, 145-2C11; BioLegend) were probed with fluorescently labeled antibodies for 20 min at 4°C in FACS buffer (0.5% BSA, 1 mM sodium-azide, and 2 mM EDTA in PBS) supplemented with Brilliant Stain Buffer Plus (BD Biosciences). Immune cells in lungs and spleens of wt or LAV^Nsp16-infected mice were collected 21 days postinfection.

Lungs were dissected and incubated in 2 ml R10 medium (RPMI, 1640, 10% fetal calf serum, FCS, 2 mM L-Glutamine, 10 mM HEPES, 50 µM β-mercaptoethanol, and 1% streptomycin/penicillin) supplemented with collagenase D (500 units) and DNase I (160 units) for 45 min at 37°C. Subsequently, tissues were mashed trough a 70 µm cell strainer (Greiner Bio-One) and erythrocytes were removed by ammonium-chloride-potassium lysis (Gibco). After an additional filtration step (70 µm), samples were subjected to automated cell counting (Luna Automated Cell Counter, Logos Biosystems).

To identify Spike (S)-specific T cells, lung and spleen-derived lymphocytes were probed with APC-labeled VNFNFNGL-loaded H-2Kb Dextramer (1:20, Immudex) in FACS buffer for 20 minutes at 4°C. For phenotypic analysis, cells were subsequently probed with anti-CD69-BV421 (H1.2F3, BioLegend), anti-CD4-BV605 (clone RM4–5, BioLegend), anti-CD8-BV711 (clone 53-6.7, BioLegend), anti-CD49a-BV786 (clone Ha31/8, BD Biosciences), anti-CD45-FITC (clone 104, BioLegend), anti-CD103-PE (clone 2E7, Invitrogen), anti-CD127-PE/Dazzl 594 (clone A7R34, BioLegend), anti-CD44-PE/Cy5 (clone IM7, BioLegend), anti-CD11a-PerCP-eFluor 710 (clone M17/4, Invitrogen), anti-KLRG1-PE/Cy7 (clone 2F1, Invitrogen), and Fixable Viability Dye eFluor 780 (ThermoFischer) for 20 min at 4°C.

For intracellular cytokine staining, T cells were restimulated in 200 µl R10 medium supplemented with monensin (2 µM), anti-CD28 (clone 37.51; eBioscience), anti-CD107a-FITC (clone 1D4B; BD Bioscience), as well as the S peptides VNFNFNGL (5 µg/ml) and VTWFHAIHVSGTNGT (5 µg/ml) for 16 hours at 37°C. Subsequently, cells were probed with anti-CD8a-Pacific Blue (clone 53-6.7; BioLegend), anti-CD4-PerCP eFluor 710 (clone RM4–5; eBioscience), and Fixable Viability Dye eFlour 780 (Thermo) for 20 min at 4°C. After fixation with 2% PFA for 20 min at 4°C, cells were permeabilized using 1% saponin and probed with anti-CD16/32 (Invitrogen), anti-IL2-APC (clone JES6-5H4, BioLegend), anti-TNFα-PE-Cy7 (clone MPG-XT22, BioLegend), and anti-IFNγ-PE (clone XMG1.2, BioLegend) for 30 min at 4°C. To identify S-specific B cells, biotin-labeled S (500 ng/sample; 130-127-682, Miltenyi) was coupled to either streptavidin-BV785 or streptavidin-PEVio615 for 15 min at room temperature (RT). Lung and spleen-derived lymphocytes were then stained with both S-complexes and anti-CD69-BV421 (clone H1.2F3, BioLegend), anti-IgG-BV605 (poly4053, BioLegend), anti-CD19-BV650 (6D5, BioLegend), anti-B220-BV711 (clone RA3-6B2, BioLegend), anti-IgA-FITC (polyclonal, Bethyl Laboratories), anti-GL7-PE (clone GL7, BioLegend), anti-CD138-PE/Cy5 (clone 281–2, BioLegend), anti-IgD-PE/Cy7 (clone 11-26c.2a, BioLegend), anti-IgM-PE/Cy7 (clone RMM-1, BioLegend), anti-CD95-APC (clone SA367H8, BioLegend), anti-CD45.2-Alexa Fluor 700 (clone 104, BioLegend), Fixable Viability Dye eFluor 780 (eBioscience), anti-CD11a-PerCP-eFluor 710 (clone M17/4, Invitrogen) in FACS-Buffer (PBS, 0.5% BSA, 1 mM sodium-azide, and 2 mM EDTA) for 30 minutes at 4°C. All cells were fixed with 2% PFA and analyzed using Attune NxT (ThermoFisher) or Northern Light (Cytek) flow cytometer (S3 and S5 Figs).

## Spike-specific antibody detection

For S-specific antibody detection by ELISA, white 96-well F-bottom plates were coated with 100 ng S protein (Bio-Techne) in coating buffer (8.4 g/l NaHCO$_3$, 3.56 g/l Na$_2$CO$_3$) at 4°C over night. Plates were blocked with 0.1% BSA in PBS-Tween for one hour at RT prior to incubation with samples. Serum samples were diluted 1:100 and BAL samples were diluted 1:30 in PBS-Tween/ 0.1% BSA and incubated for one hour at RT. Next, plates were washed and probed with HRP-coupled anti-mouse IgG (1:3000, poly4053, BioLegend) or anti-mouse IgA (1:5000, polyclonal, Bethyl) in PBS-Tween/ 0.1% BSA for one hour at RT. After addition of 50 µl ECL solution, HRP luminescence was quantified on a plate reader (VICTOR X5, PerkinElmer).

## Pseudotyped virus neutralization assay

To assess the neutralizing capacity of S-specific antibodies, serum and BAL samples were incubated with SIV-luciferase reporter virus pseudotyped with SARS-CoV-2 S (B.1 variant). Viruses were produced as described before and incubated with serial serum and BAL dilutions in DMEM supplemented with 10% FCS, 1% Penicillin/Streptomycin, and 1% Glutamax (Gibco) for one hour at 37°C. [30] Next, HEK293-T cells stably expressing human ACE2 were incubated in 96-well F-bottom plates with the antibody-particle mixture at 37°C. After 48 hours, cell supernatant was removed and 100 µl Bright

Glo lysis buffer (Promega) was added. After 15 min at 37°C, 25 μl Bright Glo substrate (Promega) was added and luciferase activity was quantified on a plate reader (VICTOR X4, PerkinElmer).

### Tissue damage

To asses protein concentration in BAL as surrogate for tissue damage, BAL was heat-inactivated at 65°C for 45 min. Protein concentration was determined via BCA assay (Pierce) and analyzed on a plate reader (VICTOR X4, PerkinElmer).

### Immunoblot analysis

Brain and lung tissue samples were homogenized in Glo Lysis Buffer (Promega Corporation). Protein concentration in lysate was determined by BCA assay (Pierce). Normalized amounts of lysate were separated by SDS-PAGE and transferred onto Immobilon-P PVDF membranes. After blocking with 5% non-fat dry milk, membranes were probed with primary antibodies targeting ACE2 (clone SN0754, Novus Biologicals) and GAPDH (clone 14C10, Cell Signaling). Secondary antibody staining was performed with an anti-rabbit HRP-coupled antibody (clone 7074, Cell Signaling).

### Histopathological analysis

Lungs of euthanized mice were filled with 4% PFA in PBS to ensure fixation and prevent the collapse of pulmonary airways. After resection, tissue was submerged in 4% PFA for 16–24 hours at 4°C. Fixed lungs were embedded in paraffin and 5 μm tissue sections were prepared using a microtome. Tissue slides were stained with hematoxylin and eosin (H&E). Slides were scanned by S210 digital slide scanner from Hamamatsu (0.23 μm/pixel; S210, Hamamatsu, Japan). Pictures were generated using the Case Viewer software (3DHistech, Hungary). H&E score was determined by assessing perivascular, peribronchial, intrabronchial, or parenchymal inflammatory events (single score 0–4; total score 0–16).

### Murine and human ACE2 structure analysis

Structural analysis was based on the crystal structure of SARS-CoV-2 RBD in complex with hACE2 (PDB: 6LZG). [28] To rationalize the differences between mACE2 and hACE2, the four exchanges Q24N, D30N, K31N, and H34Q were introduced in hACE2 with SwissModel. [44] PISA was used for the analysis of intermolecular interactions and RasMol for structure visualization. [45, 46]

## Supporting information

**S1 Table. Detailed statistical analysis of Figs 1–8. (xlsx).**
(XLSX)

**S1 Raw Images. Original images depicted in Figs 2D, 6H and 8A.**
(ZIP)

**S1 Fig. (A) Correlation between viral genome copies and IFN-ß transcripts in brain tissue. WT virus or LAV genome copies as well as IFN-ß transcripts were quantified in the brain of K18-hACE2 (from Fig1) or chACE2 mice (from Fig 3) at 5–7 dpi by RT-qPCR. (B)** Viral genome copies, IFN-ß transcripts, and maximal clinical scores are shown for individual animals. Fold change of IFN-ß transcripts in infected animals compared to naive animals is shown.
(EPS)

**S2 Fig. At 4 dpi of chACE2 and K18-hACE2 mice, wt virus was isolated from the lung and sequenced.** The viral S protein sequence was analyzed and plotted against input virus sequence (WT) as well as the sequence of the mouse-adapted strain MA20. Red = AA residues mutated in MA20. Bold = AA residues crucial for ACE2-RBD interaction.
(EPS)

**S3 Fig. Analysis of infectious virus isolate by immunoflourescence. Caco-2 cells were infected in a 96-well format with serial dilutions of viral isolates from BAL of aged animals aquired at 2 dpi.** Infectious titers were determined by infection an immunofluorescence-based infectivity assay probing for viral spike (S) protein. Exemplary wells incubated with 1:10 BAL dilutions are shown.
(EPS)

**S4 Fig. Gating strategy for flow cytometry analysis of BAL cells.**
(EPS)

**S5 Fig. Gating strategy for flow cytometry analysis of B cells.**
(TIF)

**S6 Fig. Gating strategy for flow cytometry analysis of T cells.**
(TIF)

**S7 Fig. Gating strategy for flow cytometry analysis of intracellular cytokine staining (ICS).**
(TIF)

## Acknowledgments

We thank Anna Schmidt, Harald zur Hausen Institute of Virology, Uniklinikum Erlangen, for technical assistance, Nada Cordasic, Department of Internal Medicine 4, Uniklinikum Erlangen, for veterinary assistance and animal welfare, and Jannik Wagner, Harald zur Hausen Institute of Virology, Uniklinikum Erlangen, for providing K18-hACE2 mice.

## Author contributions

**Conceptualization:** Thomas Gramberg.

**Data curation:** Alina Russ, Vera Viherlehto, Heinrich Sticht, Thomas Gramberg.

**Formal analysis:** Vera Viherlehto, Sabine Wittmann, Pascal Irrgang, Arne Cordsmeier, Armin Ensser.

**Funding acquisition:** Armin Ensser, Ralf J. Rieker, Matthias Tenbusch, Thomas H. Winkler, Thomas Gramberg.

**Investigation:** Alina Russ, Vera Viherlehto, Stefanie Brey, Sabine Wittmann, Pascal Irrgang, Arne Cordsmeier, Carol Geppert, Heinrich Sticht, Matthias Tenbusch, Thomas H. Winkler, Thomas Gramberg.

**Methodology:** Alina Russ, Vera Viherlehto, Stefanie Brey, Natascha Leicht, Carol Geppert, Heinrich Sticht, Thomas H. Winkler, Thomas Gramberg.

**Resources:** Armin Ensser, Matthias Tenbusch, Thomas H. Winkler.

**Supervision:** Thomas H. Winkler, Thomas Gramberg.

**Validation:** Thomas Gramberg.

**Visualization:** Alina Russ, Heinrich Sticht.

**Writing – original draft:** Thomas H. Winkler, Thomas Gramberg.

**Writing – review & editing:** Vera Viherlehto, Ralf J. Rieker, Heinrich Sticht, Matthias Tenbusch, Thomas H. Winkler, Thomas Gramberg.

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
