## [Decision Letter · Decision Letter 0]

6 Nov 2025

Live attenuated vaccination protects aged chACE2 mice from severe SARS-CoV-2 pathogenicity in vivo

PLOS Pathogens

Dear Dr. Gramberg,

Thank you for submitting your manuscript to PLOS Pathogens. After careful consideration, we feel that it has merit but does not fully meet PLOS Pathogens's publication criteria as it currently stands. Therefore, we invite you to submit a revised version of the manuscript that addresses the points raised during the review process. The reviewers made quite some recommendations to improve the manuscript. Please review these and address especially the most critical comments made.

We look forward to receiving your revised manuscript.

Kind regards,

Bart L. Haagmans

Academic Editor

PLOS Pathogens

Alexander Gorbalenya

Section Editor

PLOS Pathogens

Editor-in-Chief

PLOS Pathogens

orcid.org/0000-0003-2946-9497

Editor-in-Chief

PLOS Pathogens

orcid.org/0000-0002-7699-2064

**Additional Editor Comments:**

The reviewers made quite some recommendations to improve the manuscript. Please review these and address especially the most critical comments made.

**Journal Requirements:**

- ® on pages: 11, 12, and 13

- TM on pages: 13, 14, and 15.

5) We note that your Data Availability Statement is currently as follows: "All relevant data are within the manuscript and its Supporting Information files.". Please confirm at this time whether or not your submission contains all raw data required to replicate the results of your study. Authors must share the “minimal data set” for their submission. PLOS defines the minimal data set to consist of the data required to replicate all study findings reported in the article, as well as related metadata and methods (https://journals.plos.org/plosone/s/data-availability#loc-minimal-data-set-definition).

**Reviewers' Comments:**

Reviewer's Responses to Questions

**Part I - Summary**

Reviewer #1: The manuscript "Live attenuated vaccination protects aged chACE2 mice from severe SARS-Co-2 pathogenicity in vivo” by Russ et al., introduces a novel murine model (chACE2) that expresses a human-mouse chimeric ACE2 receptor at physiological levels, enabling SARS-CoV-2 infection without the neuroinvasive lethality seen in previous models. Using this model, the authors evaluate a live attenuated vaccine candidate lacking Nsp16 methyltransferase activity (LAVNsp16), demonstrating its ability to elicits mucosal and systemic immunity and confer protection against severe disease in aged mice challenged with both homologous and heterologous SARS-CoV-2 variants.

The manuscript is well-conceived and contributes to the field. With revisions addressing the points, it will be suitable for publication in Plos Pathogens.

Reviewer #2: The manuscript by Russ et al. describes the generation of a SARS-CoV-2 animal model consisting in knock-in C57BL/6 mice, in which the murine ACE2 has been replaced by a human-mouse chimeric ACE2 (chACE2), using CRISPR-Cas9. In this chimeric receptor, four residues at positions 24, 30, 31, and 34 of the murine ACE2 were substituted with their human counterparts to enhance RBD binding. The knock-in strategy led to physiological levels of chACE2 in terms or abundance and tissue distribution, avoiding brain overexpression—an advantage over the transgenic K18-hACE2 mouse model.

Other studies have also reported similar mouse models with more physiological expression of the viral receptor. In this manuscript, aged chACE2 mice were used to evaluate the efficacy of a vaccine candidate, which is a novel and highly relevant aspect, given that aged individuals are more vulnerable to SARS-CoV-2 pathology and have weaker immune responses to infection and vaccination.

As the vaccine candidate, a live-attenuated virus (LAVnsp16) previously engineered by the authors, was used.

The manuscript provides valuable data on a new mouse model that mimics human infection, showing low pathogenicity in young animals and aggravated disease in older animals (42 week-old). However, several concerns regarding the conclusions and the experimental approach should be addressed.

Reviewer #3: The manuscript of Russ et al. describes an interesting novel chimericACE2 and its utilization for vaccine evaluation with a live attenuated vaccine. Despite overcoming some very relevant limitations of the K18-hACE2 mouse model with the new model development, findings on in-vivo studies need further verification due to a limited sample size of comparable conditions. Furthermore, the analysis of gained in-vivo samples remains rather superficial. Overall the study would grow to its full potential by further in-depth analysis of samples and confirmation of findings. Since the development and characterization of relevant in-vivo models is still a big need, the value of the manuscript is given. To convince readers and scientific community of the value of the model it would yet need further confirmation.

**Part II – Major Issues: Key Experiments Required for Acceptance**

Please use this section to detail the key new experiments or modifications of existing experiments that should be absolutely required to validate study conclusions.required to validate study conclusions.required to validate study conclusions.required to validate study conclusions.

Reviewer #1: The results section presents compelling data supporting the utility of the chACE2 model and the immunogenicity and safety of LAVNsp16. However, several key aspects mentioned in Part III require new experiments, clarification or additional information: (i) infectious virus titration, (ii) histopathological assessment of tissue damage, and (iii) assessment of neutralizing antibody responses.

Reviewer #2: 1. The chimeric receptor (chACE2) is not fully human, as only four residues involved in S-RBD binding were replaced in the murine ACE2. This may result in suboptimal interaction with the S protein, potentially driving to adaptive evolution to increase viral fitness. Sequencing of SARS-CoV-2 recovered from infected chACE2 mice should be performed to exclude the generation of a chACE2-adapted virus. This could limit the model´s utility for pathogenesis studies, though it would still be valid for evaluating vaccines or antivirals.

2. To study the viral growth in the chACE2 mouse model, viral RNA was quantified by RT-qPCR (Figs. 3, 6 and 8). To enhance physiological relevance, infectious virus titers should be provided.

3. FIg. 4. Are the differences in immune cell infiltration between wt and LAVnsp16 infection statistically significant?

4. No significant differences in infiltrating cells and ISG expression are observed between LAVnsp16 and SARS-CoV-2 wt at 4 and 7 dpi in K18-hACE2 mice (Fig. 4). Similarly, LAVnsp16 and SARS-CoV-2 wt also induced comparable cell infiltration and tissue damage (Fig. 7) in aged chACE2 mice. These results suggest residual virulence in LAVnsp16 and additional attenuation may be needed for further vaccine development. Could the authors discuss this point?

5. Fig. 5B. Please discuss the observation that the anti-S IgG response induced by LAVnsp16 in BAL is not superior to naive mice, despite intranasal administration and high LAV RNA levels in BAL and lungs.

6. Fig. 6G. Different ACE-2 protein bands patterns are observed between young and aged mice in both wt B6 and in chACE2 models. Please discuss.

7. Fig. 1D and Fig. 7F. Protein concentration in BAL fluid is used as a surrogate for tissue damage. This indirect measure should be complemented or replaced by histopathological analysis to confirm vaccine attenuation in both in young and aged mice.

8. Fig. 8B and 8D show that the heterologous Delta variant challenge in non-immunized animals does not cause significant weight loss or mortality, suggesting lower replication efficiency in chACE2 mice compared to B.1 variant. The conclusion that immunization potently protects against heterologous and homologous SARS-CoV-2 replication (line 283) should be revised.

Reviewer #3: A further characterization of actual induced pathology or lack thereof would benefit this initial model characterization. Currently analysis focusses on viral genome quantification (no infectious virus), mortality and immune response. Especially if consequences for elderly shall be mimicked and side effects of LAV shall be evaluated, further analysis of infectious virus titers and pathological analysis (at least viral antigen quantification in respiratory tissue) would be of added value. Also a comparison of infected cell types vs human cases would be interesting if this manuscript is the first who describes this model. Therefore I recommend follow-up analysis of tissues in regards of histopathological analysis.

For characterization of reduced pathology of the LAV in K18-ACE2 mice the authors still observe high viral genome in the brain of 50% of animals. Yet mortality is reduced. Can the authors explain that? Also here, histopathology or infectious virus titration would be interesting to explain observations.

ll.214: The authors mention a perspective of intranasal vaccination of LAV. Where IgA levels checked as well? Antibodies are mentioned in the methods section, but mucosal immunity is not in the results. Would be very interesting to look at that aspect.

ll.245: The authors state that viral genome only slightly differs between old and young inoculated mice. However, disease severity seems very different. Also here, further analysis would benefit the overall message and conclusions of the paper.

Fig.3D: There is still a lot of viral genome (maybe no infectious virus?) in the lungs of vaccinated animals. How can you be sure, if no data of histopathology are available, that your vaccine is really safe?

ll.268: Can the authors explain why Delta was chosen as heterologous challenge? This variant is not of relevance anymore and the authors mention immune-escape of Omicron variants and therefore limitations of spike mRNA vaccines in the intro. Wouldn’t it have been a better proof of concept to use an Omicron variant as heterologous challenge to really prove an added value of your vaccine?

Biggest issue of the manuscript is that data of young vs old chACE2 mice are barely comparable. However, this distinct phenotype remains one of the main observations of the manuscript. One issue that young mice were sacrificed on day 2,4 and 7 (Fig 3). However, old mice were then chosen to be sacrificed on day 5. Figure 6 compares young vs old, yet looking at 2 different dpi (4 vs 5). In addition, 2 animals reached humane endpoints even earlier on (2 and 3 dpi). Despite of n=4, only 3 datapoints are shown for aged animals. Meaning that we look at data from 3 different timepoints of a single animal comparing to a group of 4 from yet another timepoint. This makes comparison and conclusions impossible, which is relevant for figure 6, 7 and 8. In your figure 3D you indicate the relevance of viral kinetics and how significant changes are over just 2 days. Therefore, none of the conclusion drawn from these data are valid due to n=1.

**Part III – Minor Issues: Editorial and Data Presentation Modifications**

Reviewer #1: The manuscript is well-conceived and contributes to the field. With revisions addressing the points, it will be suitable for publication in Plos Pathogens. The following points need to be addressed by the authors:

(i) Abstract: it clearly identifies the problem (lack of physiologically relevant mouse models for SARS-CoV-2, especially in aged populations) and introduces a novel solution (the chACE2 mouse model). The structure of the abstract is logical, moving from model development to vaccine testing and outcomes. Areas for improvement:

- Overuse of general statements. Phrases like “Strikingly…” and “Together, we show…” are more appropriate for the discussion; the abstract should remain objective and concise.

(ii) Introduction: it provides an overview of the COVID-19 vaccine landscape; it identifies the limitations of current murine models (the need for improved systems to study age-related SARS-CoV-2 pathology); and it introduces the rationale for targeting Nsp16 and the development LAVNsp16. However, some aspects can be improved:

- Clarify claims such as “problems of vaccine strategies targeting solely the S protein became visible” is vague. Specify what problems (immune escape, reduced efficacy, …).

- “huge number of sub lineages” is informal. Consider “extensive diversification of Omicron sublineages”.

- The introduction refers on age-related immune decline but could benefit from a mechanistic discussion such as immunosenescence or impaired IFN signaling (1-2 sentences).

- What advantages does the chimera offer vs full human ACE2? Is this discussed in the Discussion section? Not clear in the introduction as “model justification”.

(iii) Results: this section presents promising data on the attenuation and immunogenicity of LAVNsp16 and the utility of the chACE2 model. However, the conclusions would be strengthened by addressing the limitations mentioned below.

- The statement in lines 131-132 is not entirely accurate, as wild-type mice have been shown to be susceptible to infection with certain SARS-CoV-2 variants, including B.1.351/Beta and BA.1.1/Omicron, as reported by Tarrés-Freixas et al. (Front Microbiol. 2022 May 4;13:840757. doi: 10.3389/fmicb.2022.840757. eCollection 2022).

- Line 137: please replace “SARS-CoV-1” with “SARS-CoV”, as the correct nomenclature for the virus identified in 2002-2003 is “SARS-CoV” (https://ictv.global/report/chapter/coronaviridae/coronaviridae/betacoronavirus)

- Line 141: please replace “qRT-PCR” with the correct terminology “RT-qPCR”, which is the standard abbreviation for revers transcription quantitative PCR.

- Line 142-143: the statement regarding enhanced tissue damaged observed in the BAL is misleading, as BAL is a fluid sample. While the figure legend notes that protein concentration in BAL was used as a surrogate marker for tissue injury, this rationale should also be explicitly stated in the results section. Additionally, it would be important to clarify why histopathological analysis of lung tissue was not performed to directly evaluate tissue damage.

- Only viral RNA quantification has been performed; was infectious virus titration (e.g., plaque assay or TCID50) conducted to confirm the presence of replication-competent virus?

- Figure 1E: is there any data available on the progression of disease in this animal model beyond 7 dpi with LAVNsp16? Specifically, could the onset or severity of disease be delayed?

- Although the authors demonstrate that chACE2 mice support replication of both wt SARS-CoV-2 and the attenuated strain (Line 179), with clear differences in pathogenicity (absence of neuroinvasion and brain viral RNA), the authors rely on viral RNA quantification. It is not clear whether infectious virus titration was performed which is critical to confirm active replication and/or attenuation.

- Figure 3E interpretation: the absence of viral RNA in the brain is crucial, but histopathological confirmation would be valuable to rule out subclinical neuroinflammation.

- The authors provide flow cytometry data showing innate immune cell infiltration and RT-qPCR data supporting local immune activation, but this reviewer questions the rationale for using RT-qPCR to assess IFNbeta and ISGs expression instead of flow cytometry to measure protein, which would provide more direct evidence of cytokine production and cellular activation.

- The authors report anti-S IgG levels (Line 211-234) but do not assess neutralizing capacity, which is the gold standard for measuring neutralizing antibodies, thus, the ability to block virus infection.

- Also, for aged mice (Lines 235-263) histological analysis of lung will support the claim of reduced tissue damage in those infected with LAVNsp16.

- The reduced ISG induction and immune cell infiltration are consistent with attenuation. However, age-related differences in baseline immune status should be discussed.

- Again, in the section “homologous or heterologous SARS-CoV-2 challenge” (Lines 265-284): viral replication in BAL and lungs was assessed via RT-qPCR and no infectious titers are provided. Quantifying replication-competent virus is essential to validate vaccine efficacy.

- The Delta variant challenge is described as resulting in no disease within four days, suggesting slower replication, but this interpretation requires caution. Longer periods post-challenge (7-10dpi) would be necessary to fully assess pathogenicity and vaccine protection.

(iv) Discussion:

- While live attenuated vaccines are known for their efficacy, the manuscript should more explicitly address the limitations of this approach, particularly in immunocompromised populations, including the elderly. This is especially relevant given the focus on aged mice.

- The authors report strong mucosa and systemic immunity, but further elaboration on the specific immune parameters measured would be beneficial.

- The manuscript would benefit from a more detailed discussion of the heterologous variants used, including their relevance to circulating strains and implications for cross-protection.

Overall assessment: the results section presents compelling data supporting the utility of the chACE2 model and the immunogenicity and safety of LAVNsp16. However, several key aspects mentioned above require clarification or additional information: (i) infectious virus titration, (ii) histopathological assessment of tissue damage, and (iii) assessment of neutralizing antibody responses.

(v) Materials and methods:

- This section would benefit from a thorough revision and the inclusion of more detailed and specific methodological information to improve clarity and reproducibility. For instance, the description in lines 399-402 is difficult to follow: “Eight-week-old K18-hACE2 mice (B6.Cg-Tg(K18-ACE2)2Prlmn/J) were infected with 1000 plaque-forming units (PFU) of virus in 30 μl PBS. Young (8 to 12 weeks) and aged (40 to 42 weeks) chACE2 mice (C57BL/6-Ace2-tm1(ACE2*1)-TW) received 7 x104 PFU in 30 μl PBS”. The protocol should be described more precisely, it is confusing.

(vi) Figures:

- Figure 1 legend: please specific the virus more clearly by replacing “1,000PFU of wt” with “1,000 PFU of SARS-CoV-2 wt”.

(vii) Revise grammar and style, for example:

- “did not result in diseases” should be revised to “did not result in clinical disease”.

- “suggesting rather slow replication kinetics” consider rephrasing to “suggesting delayed replication kinetics”.

- or, “Fig. 3. Chimeric ACE2 supports SARS-CoV-2…” replace it by “Fig. 3. Chimeric ACE2 mice supports SARS-CoV-2 wt ..:”

I recommend major revision to address these points and strengthen the manuscript’s scientific rigor.

Reviewer #2: 1. Line 216. “Corelate” should be “correlate”

2. Fig. 6C and 6D are not referenced in the text.

3. Growth of SARS-CoV-2 wt and LAV in nasal turbinates is not discussed.

4. Fig. 6H. For clarity and consistency, “huACE2” should be “chACE2” and “B6” should be “mACE2”.

5. Fig. 8E and 8F. Lines 280-281. To conclude suppression to background levels in LAV-immunized and non-challenge mice, samples from mock-immunized and non-challenge controls should be included.

6. Line 287. “live attenuate” should be “live attenuated”

Reviewer #3: Title: I would recommend to write out chACE2

Ll.44: Throughout the manuscript authors use the term “long-term consequences” and refer to an immunological response upon vaccination. To me the use of this term is misleading since it implies rather long-term effects as observed for Long COVID. I would recommend re-wording.

ll. 289: The authors mention that other chimericACE2 models were established previously (Ref29-32). It would be good to state the novelty of their own model here. What is the added value/difference/unique aspect, compared to others?

ll.633 vs 660: If K18hACE2 mice should be compared with the chACE2 mice, why were different inoculation doses used?

Throughout the whole manuscript the authors refer to viral titers. Please change to viral genome (titers), since readers can otherwise confuse it with life virus data.

ll. 639: A protein measurement of BAL is used as surrogate for tissue damage. Is that method been published before and are there actual proven correlations with damage?

ll.668: Panel E is not referenced in this caption.

Fig.8: It is indeed surprising to see (as authors also mention in the discussion) that almost no changes are observed during an infection with Delta. Also here, pathology and further downstream analysis could support explanations for this observation.

Figures of the supplement were not referenced to in the main text.

PLOS authors have the option to publish the peer review history of their article (what does this mean?). If published, this will include your full peer review and any attached files.). If published, this will include your full peer review and any attached files.). If published, this will include your full peer review and any attached files.). If published, this will include your full peer review and any attached files.

...

Reviewer #1: No

Reviewer #2: No

Reviewer #3: No

**Figure resubmission:**

**Reproducibility:**



---

## [Decision Letter · Decision Letter 1]

10 Apr 2026

Dear Prof Gramberg,

We are pleased to inform you that your manuscript 'Live attenuated vaccination protects aged chimeric ACE2 mice from severe SARS-CoV-2 pathogenicity in vivo' has been provisionally accepted for publication in PLOS Pathogens.

Best regards,

Bart L. Haagmans

Academic Editor

PLOS Pathogens

Alexander Gorbalenya

Section Editor

PLOS Pathogens

Sumita Bhaduri-McIntosh

Editor-in-Chief

PLOS Pathogens

orcid.org/0000-0003-2946-9497

Michael Malim

Editor-in-Chief

PLOS Pathogens

orcid.org/0000-0002-7699-2064

Reviewer Comments (if any, and for reference):

Reviewer's Responses to Questions

**Part I - Summary**

Reviewer #1: (No Response)

Reviewer #2: The authors have satisfactorily addressed most of this reviewer’s comments and concerns. Additional experimental data and clarifications have been included, which strengthen the relevance of the conclusions.

The manuscript has been significantly improved and is now acceptable for publication.

Reviewer #3: The authors addressed all revisions and comments satisfactory.

I would only recommend to consider adding the section and thoughts about IgA and mucosal immunization into a perspective paragraph of the discussion.

**Part II – Major Issues: Key Experiments Required for Acceptance**

Please use this section to detail the key new experiments or modifications of existing experiments that should be absolutely required to validate study conclusions.required to validate study conclusions.required to validate study conclusions.required to validate study conclusions.

Reviewer #1: (No Response)

Reviewer #2: No major issues are identified in this revised version of the manuscript.

Reviewer #3: non

**Part III – Minor Issues: Editorial and Data Presentation Modifications**

Reviewer #1: (No Response)

Reviewer #2: No minor issues are identified in this revised version of the manuscript.

Reviewer #3: Authors may consider to add the section and thoughts about IgA and mucosal immunization into a perspective paragraph of the discussion.

PLOS authors have the option to publish the peer review history of their article (what does this mean?). If published, this will include your full peer review and any attached files.). If published, this will include your full peer review and any attached files.). If published, this will include your full peer review and any attached files.). If published, this will include your full peer review and any attached files.

...

Reviewer #1: No

Reviewer #2: No

Reviewer #3: **Yes:** Melanie RissmannMelanie RissmannMelanie RissmannMelanie Rissmann

---

## [Editor Report · Acceptance letter]

Dear Prof Gramberg,

We are delighted to inform you that your manuscript, "Live attenuated vaccination protects aged chimeric ACE2 mice from severe SARS-CoV-2 pathogenicity in vivo," has been formally accepted for publication in PLOS Pathogens.

Best regards,

Sumita Bhaduri-McIntosh

Editor-in-Chief

PLOS Pathogens

orcid.org/0000-0003-2946-9497

Michael Malim

Editor-in-Chief

PLOS Pathogens

orcid.org/0000-0002-7699-2064